# Generalized Shape Metrics on Neural Representations

**Alex H. Williams**
Statistics Department
Stanford University
ahwillia@stanford.edu

**Erin Kunz**
Electrical Engineering Department
Stanford University
ekunz@stanford.edu

**Simon Kornblith**
Google Research, Toronto
skornblith@google.com

**Scott W. Linderman**
Statistics Department
Stanford University
scott.linderman@stanford.edu

## Abstract

Understanding the operation of biological and artificial networks remains a difficult and important challenge. To identify general principles, researchers are increasingly interested in surveying large collections of networks that are trained on, or biologically adapted to, similar tasks. A standardized set of analysis tools is now needed to identify how network-level covariates—such as architecture, anatomical brain region, and model organism—impact neural representations (hidden layer activations). Here, we provide a rigorous foundation for these analyses by defining a broad family of metric spaces that quantify representational dissimilarity. Using this framework, we modify existing representational similarity measures based on canonical correlation analysis and centered kernel alignment to satisfy the triangle inequality, formulate a novel metric that respects the inductive biases in convolutional layers, and identify approximate Euclidean embeddings that enable network representations to be incorporated into essentially any off-the-shelf machine learning method. We demonstrate these methods on large-scale datasets from biology (Allen Institute Brain Observatory) and deep learning (NAS-Bench-101). In doing so, we identify relationships between neural representations that are interpretable in terms of anatomical features and model performance.

## 1 Introduction

The extent to which different deep networks or neurobiological systems use equivalent representations in support of similar task demands is a topic of persistent interest in machine learning and neuroscience [1–3]. Several methods including linear regression [4, 5], canonical correlation analysis (CCA; [6, 7]), representational similarity analysis (RSA; [8]), and centered kernel alignment (CKA; [9]) have been used to quantify the similarity of hidden layer activation patterns. These measures are often interpreted on an ordinal scale and are employed to compare a limited number of networks—e.g., they can indicate whether networks $A$ and $B$ are more or less similar than networks $A$ and $C$. While these comparisons have yielded many insights [4–12], the underlying methodologies have not been extended to systematic analyses spanning thousands of networks.

To unify existing approaches and enable more sophisticated analyses, we draw on ideas from *statistical shape analysis* [13–15] to develop dissimilarity measures that are proper metrics—i.e., measures that are symmetric and respect the triangle inequality. This enables several off-the-shelf methods with theoretical guarantees for classification (e.g. k-nearest neighbors, [16]) and clustering (e.g. hierarchical clustering [17]). Existing similarity measures can violate the triangle inequality, which complicates these downstream analyses [18–20]. However, we show that existing dissimilarity

35th Conference on Neural Information Processing Systems (NeurIPS 2021).

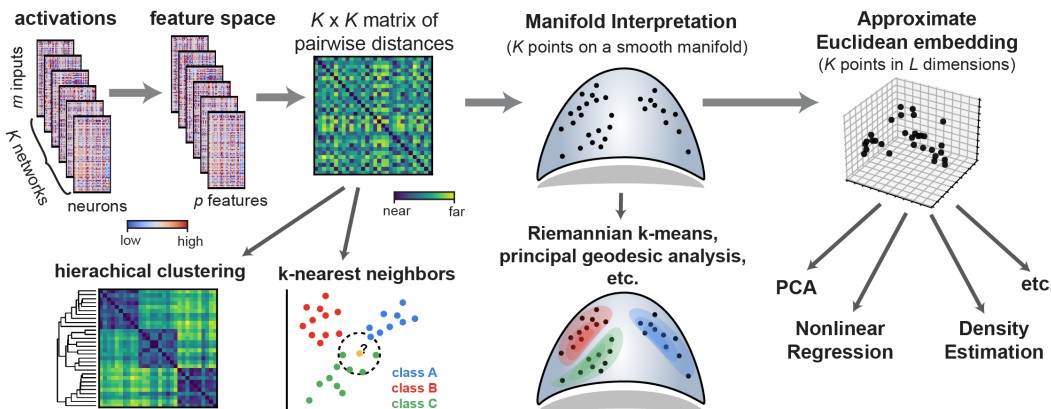

Figure 1: Machine learning workflows enabled by generalized shape metrics.

measures can often be modified to satisfy the triangle inequality and viewed as special cases of the framework we outline. We also describe novel metrics within this broader family that are specialized to convolutional layers and have appealing properties for analyzing artificial networks.

Moreover, we show empirically that these metric spaces on neural representations can be embedded with low distortion into Euclidean spaces, enabling an even broader variety of previously unconsidered supervised and unsupervised analyses. For example, we can use neural representations as the inputs to linear or nonlinear regression models. We demonstrate this approach on neural representations in mouse visual cortex (Allen Brain Observatory; [21]) in order to predict each brain region's anatomical hierarchy from its pattern of visual responses—i.e., predicting a feature of brain structure from function. We demonstrate a similar approach to analyze hidden layer representations in a database of 432K deep artificial networks (NAS-Bench-101; [22]) and find a surprising degree of correlation between early and deep layer representations.

Overall, we provide a theoretical grounding which explains why existing representational similarity measures are useful: they are often close to metric spaces, and can be modified to fulfill metric space axioms precisely. Further, we draw new conceptual connections between analyses of neural representations and established research areas [15, 23], utilize these insights to propose novel metrics, and demonstrate a general-purpose machine learning workflow that scales to datasets with thousands of networks.

## 2 Methods

This section outlines several workflows (Fig. 1) to analyze representations across large collections of networks. After briefly summarizing prior approaches (sec. 2.1), we cover background material on metric spaces and discuss their theoretical advantages over existing dissimilarity measures (sec. 2.2). We then present a class of metrics that capture these advantages (sec. 2.3) and cover a special case that is suited to convolutional layers (sec. 2.4). We then demonstrate the practical advantages of these methods in Section 3, and demonstrate empirically that Euclidean feature spaces can approximate the metric structure of neural representations, enabling a broad set of novel analyses.

### 2.1 Prior work and problem setup

Neural network representations are often summarized over a set of $m$ reference inputs (e.g. test set images). Let $\boldsymbol{X}_i \in \mathbb{R}^{m \times n_i}$ and $\boldsymbol{X}_j \in \mathbb{R}^{m \times n_j}$ denote the responses of two networks (with $n_i$ and $n_j$ neurons, respectively) to a collection of these inputs. Quantifying the similarity between $\boldsymbol{X}_i$ and $\boldsymbol{X}_j$ is complicated by the fact that, while the $m$ inputs are the same, there is no direct correspondence between the neurons. Even if $n_i = n_j$, the typical Frobenius inner product, $\langle \boldsymbol{X}_i, \boldsymbol{X}_j \rangle = \mathrm{Tr}[\boldsymbol{X}_i^\top \boldsymbol{X}_j]$, and metric, $\|\boldsymbol{X}_i - \boldsymbol{X}_j\| = \langle \boldsymbol{X}_i - \boldsymbol{X}_j, \boldsymbol{X}_i - \boldsymbol{X}_j \rangle^{1/2}$, fail to capture the desired notion of dissimilarity. For instance, let $\boldsymbol{\Pi}$ denote some $n \times n$ permutation matrix and let $\boldsymbol{X}_i = \boldsymbol{X}_j \boldsymbol{\Pi}$. Intuitively, we should consider $\boldsymbol{X}_i$ and $\boldsymbol{X}_j$ to be identical in this case since the ordering of neurons is arbitrary. Yet, clearly $\|\boldsymbol{X}_i - \boldsymbol{X}_j\| \neq 0$, except in very special cases.

One way to address this problem is to linearly regress over the neurons to predict $\boldsymbol{X}_i$ from $\boldsymbol{X}_j$. Then, one can use the coefficient of determination ($R^2$) as a measure of similarity [4, 5]. However, this similarity score is asymmetric—if one instead treats $\boldsymbol{X}_j$ as the dependent variable that is predicted from $\boldsymbol{X}_i$, this will result in a different $R^2$. Canonical correlation analysis (CCA; [6, 7]) and linear centered kernel alignment (linear CKA; [9, 24]) also search for linear correspondences between neurons, but have the advantage of producing symmetric scores. Representational similarity analysis (RSA; [8]) is yet another approach, which first computes an $m \times m$ matrix holding the dissimilarities between all pairs of representations for each network. These *representational dissimilarity matrices* (RDMs), are very similar to the $m \times m$ kernel matrices computed and compared by CKA. RSA traditionally quantifies the similarity between two neural networks by computing Spearman's rank correlation between their RDMs. A very recent paper by Shahbazi et al. [25], which was published while this manuscript was undergoing review, proposes to use the Riemannian metric between positive definite matrices instead of Spearman correlation. Similar to our results, this establishes a metric space that can be used to compare neural representations. Here, we leverage metric structure over *shape spaces* [13–15] instead of positive definite matrices, leading to complementary insights.

In summary, there are a diversity of methods that one can use to compare neural representations. Without a unifying theoretical framework it is unclear how to choose among them, use their outputs for downstream tasks, or generalize them to new domains.

## 2.2   Feature space mapping, metrics, and equivalence relations

Our first contribution will be to establish formal notions of distance (metrics) between neural representations. To accommodate the common scenario when the number of neurons varies across networks (i.e. when $n_i \neq n_j$), we first map the representations into a common feature space. For each set of representations, $\boldsymbol{X}_i$, we suppose there is a mapping into a $p$-dimensional feature space, $\boldsymbol{X}_i \mapsto \boldsymbol{X}_i^\phi$, where $\boldsymbol{X}_i^\phi \in \mathbb{R}^{m \times p}$. In the special case where all networks have equal size, $n_1 = n_2 = \ldots = n$, we can express the feature mapping as a single function $\phi : \mathbb{R}^{m \times n} \mapsto \mathbb{R}^{m \times p}$, so that $\boldsymbol{X}_i^\phi = \phi(\boldsymbol{X}_i)$. When networks have dissimilar sizes, we can map the representations into a common dimension using, for example, PCA [6].

Next, we seek to establish *metrics* within the feature space, which are distance functions that satisfy:

$$\text{Equivalence:} \quad d(\boldsymbol{X}_i^\phi, \boldsymbol{X}_j^\phi) = 0 \iff \boldsymbol{X}_i^\phi \sim \boldsymbol{X}_j^\phi \tag{1}$$

$$\text{Symmetry:} \quad d(\boldsymbol{X}_i^\phi, \boldsymbol{X}_j^\phi) = d(\boldsymbol{X}_j^\phi, \boldsymbol{X}_i^\phi) \tag{2}$$

$$\text{Triangle Inequality:} \quad d(\boldsymbol{X}_i^\phi, \boldsymbol{X}_j^\phi) \leq d(\boldsymbol{X}_i^\phi, \boldsymbol{X}_k^\phi) + d(\boldsymbol{X}_k^\phi, \boldsymbol{X}_j^\phi) \tag{3}$$

for all $\boldsymbol{X}_i^\phi$, $\boldsymbol{X}_j^\phi$, and $\boldsymbol{X}_k^\phi$ in the feature space. The symbol '$\sim$' denotes an *equivalence relation* between two elements. That is, the expression $\boldsymbol{X}_i^\phi \sim \boldsymbol{X}_j^\phi$ means that "$\boldsymbol{X}_i^\phi$ is equivalent to $\boldsymbol{X}_j^\phi$." Formally, distance functions satisfying Eqs. (1) to (3) define a metric over a quotient space defined by the equivalence relation and a pseudometric over $\mathbb{R}^{m \times p}$ (see Supplement A). Intuitively, by specifying different equivalence relations we can account for symmetries in network representations, such as permutations over arbitrarily labeled neurons (other options are discussed below in sec. 2.3).

Metrics quantify dissimilarity in a way that agrees with our intuitive notion of distance. For example, Eq. (2) ensures that the distance from $\boldsymbol{X}_i^\phi$ to $\boldsymbol{X}_j^\phi$ is the same as the distance from $\boldsymbol{X}_j^\phi$ to $\boldsymbol{X}_i^\phi$. Linear regression is an approach that violates this condition: the similarity measured by $R^2$ depends on which network is treated as the dependent variable.

Further, Eq. (3) ensures that distances are self-consistent in the sense that if two elements ($\boldsymbol{X}_i^\phi$ and $\boldsymbol{X}_j^\phi$) are both close to a third ($\boldsymbol{X}_k^\phi$), then they are necessarily close to each other. Many machine learning models and algorithms rely on this triangle inequality condition. For example, in clustering, it ensures that if $\boldsymbol{X}_i^\phi$ and $\boldsymbol{X}_j^\phi$ are put into the same cluster as $\boldsymbol{X}_k^\phi$, then $\boldsymbol{X}_i^\phi$ and $\boldsymbol{X}_j^\phi$ cannot be too far apart, thus implying that they too can be clustered together. Intuitively, this establishes an appealing transitive relation for clustering, which can be violated when the triangle inequality fails to hold. Existing measures based on CCA, RSA, and CKA, are symmetric, but do not satisfy the triangle inequality. By modifying these approaches to satisfy the triangle inequality, we avoid potential pitfalls and can leverage theoretical guarantees on learning in proper metric spaces [16–20].

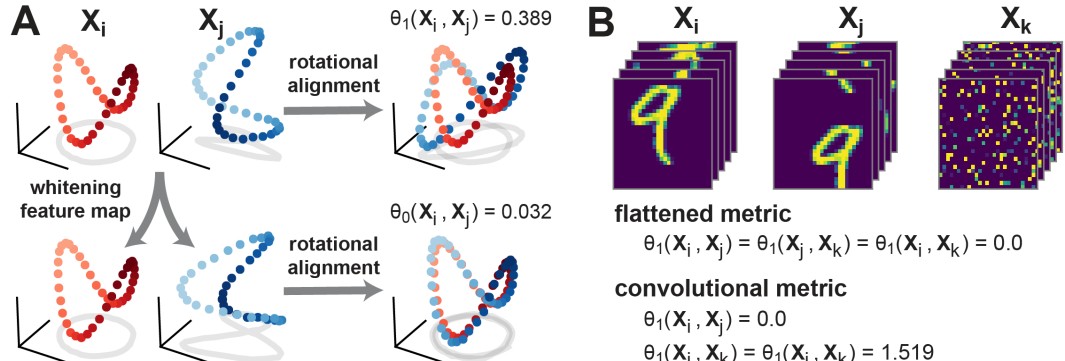

Figure 2: *(A)* Schematic illustration of metrics with rotational invariance (top), and linear invariance (bottom). Red and blue dots represent a pair of network representations $\boldsymbol{X}_i$ and $\boldsymbol{X}_j$, which correspond to $m$ points in $n$-dimensional space. *(B)* Demonstration of convolutional metric on toy data. Flattened metrics (e.g. [6, 9]) that ignore convolutional layer structure treat permuted images ($\boldsymbol{X}_k$, right) as equivalent to images with coherent spatial structure ($\boldsymbol{X}_i$ and $\boldsymbol{X}_j$, left and middle). A convolutional metric, Eq. (11), distinguishes between these cases while still treating $\boldsymbol{X}_i$ and $\boldsymbol{X}_j$ as equivalent (obeying translation invariance).

## 2.3 Generalized shape metrics and group invariance

In this section, we outline a new framework to quantify representational dissimilarity, which leverages a well-developed mathematical literature on *shape spaces* [13–15]. The key idea is to treat $\boldsymbol{X}_i^\phi \sim \boldsymbol{X}_j^\phi$ if and only if there exists a linear transformation $\boldsymbol{T}$ within a set of allowable transformations $\mathcal{G}$, such that $\boldsymbol{X}_i^\phi = \boldsymbol{X}_j^\phi \boldsymbol{T}$. Although $\mathcal{G}$ only contains linear functions, nonlinear alignments between the raw representations can be achieved when the feature mappings $\boldsymbol{X}_i \mapsto \boldsymbol{X}_i^\phi$ are chosen to be nonlinear. Much of shape analysis literature focuses on the special case where $p = n$ and $\mathcal{G}$ is the special orthogonal group $\mathcal{SO}(n) = \{\boldsymbol{R} \in \mathbb{R}^{n \times n} \mid \boldsymbol{R}^\top \boldsymbol{R} = \boldsymbol{I}, \det(\boldsymbol{R}) = 1\}$, meaning that $\boldsymbol{X}_i^\phi$ and $\boldsymbol{X}_j^\phi$ are equivalent if there is a $n$-dimensional rotation (without reflection) that relates them. Standard shape analysis further considers each $\boldsymbol{X}_i^\phi$ to be a mean-centered $((\boldsymbol{X}_i^\phi)^\top \boldsymbol{1} = \boldsymbol{0})$ and normalized $(\|\boldsymbol{X}_i^\phi\| = 1)$ version of the raw landmark locations held in $\boldsymbol{X}_i \in \mathbb{R}^{m \times n}$ (an assumption that we will relax). That is, the feature map $\phi : \mathbb{R}^{m \times n} \mapsto \mathbb{S}^{m \times n}$ transforms the raw landmarks onto the hypersphere, denoted $\mathbb{S}^{m \times n}$, of $m \times n$ matrices with unit Frobenius norm. In this context, $\boldsymbol{X}_i^\phi \in \mathbb{S}^{m \times n}$ is called a "pre-shape." By removing rotations from a pre-shape, $[\boldsymbol{X}_i^\phi] = \{\boldsymbol{S} \in \mathbb{S}^{m \times n} \mid \boldsymbol{S} \sim \boldsymbol{X}_i^\phi\}$ for pre-shape $\boldsymbol{X}_i^\phi$, we recover its "shape."

To quantify dissimilarity in neural representations, we generalize this notion of shape to include other feature mappings and alignments. The minimal distance within the feature space, after optimizing over alignments, defines a metric under suitable conditions (Fig. 2A). This results in a broad variety of *generalized shape metrics* (see also, ch. 18 of [15]), which fall into two categories as formalized by the pair of propositions below. Proofs are provided in Supplement B.

**Proposition 1.** *Let $\boldsymbol{X}_i^\phi \in \mathbb{R}^{m \times p}$, and let $\mathcal{G}$ be a group of linear isometries on $\mathbb{R}^{m \times p}$. Then,*

$$d(\boldsymbol{X}_i^\phi, \boldsymbol{X}_j^\phi) = \min_{\boldsymbol{T} \in \mathcal{G}} \|\boldsymbol{X}_i^\phi - \boldsymbol{X}_j^\phi \boldsymbol{T}\| \tag{4}$$

*defines a metric, where $\boldsymbol{X}_i^\phi \sim \boldsymbol{X}_j^\phi$ if and only if there is a $\boldsymbol{T} \in \mathcal{G}$ such that $\boldsymbol{X}_i^\phi = \boldsymbol{X}_j^\phi \boldsymbol{T}$.*

**Proposition 2.** *Let $\boldsymbol{X}_i^\phi \in \mathbb{S}^{m \times p}$, and let $\mathcal{G}$ be a group of linear isometries on $\mathbb{S}^{m \times p}$. Then,*

$$\theta(\boldsymbol{X}_i^\phi, \boldsymbol{X}_j^\phi) = \min_{\boldsymbol{T} \in \mathcal{G}} \arccos \left\langle \boldsymbol{X}_i^\phi, \boldsymbol{X}_j^\phi \boldsymbol{T} \right\rangle \tag{5}$$

*defines a metric, where $\boldsymbol{X}_i^\phi \sim \boldsymbol{X}_j^\phi$ if and only if there is a $\boldsymbol{T} \in \mathcal{G}$ such that $\boldsymbol{X}_i^\phi = \boldsymbol{X}_j^\phi \boldsymbol{T}$.*

Two key conditions appear in these propositions. First, $\mathcal{G}$ must be a *group* of functions. This means $\mathcal{G}$ is a set that contains the identity function, is closed under composition ($\boldsymbol{T}_1 \boldsymbol{T}_2 \in \mathcal{G}$ for any $\boldsymbol{T}_1 \in \mathcal{G}$ and $\boldsymbol{T}_2 \in \mathcal{G}$), and whose elements are invertible by other members of the set (if $\boldsymbol{T} \in \mathcal{G}$ then $\boldsymbol{T}^{-1} \in \mathcal{G}$).

Second, every $\boldsymbol{T} \in \mathcal{G}$ must be an *isometry*, meaning that $\|\boldsymbol{X}_i^\phi - \boldsymbol{X}_j^\phi\| = \|\boldsymbol{X}_i^\phi \boldsymbol{T} - \boldsymbol{X}_j^\phi \boldsymbol{T}\|$ for all $\boldsymbol{T} \in \mathcal{G}$ and all elements of the feature space. On $\mathbb{R}^{m \times p}$ and $\mathbb{S}^{m \times p}$, all linear isometries are orthogonal transformations. Further, the set of orthogonal transformations, $\mathcal{O}(p) = \{\boldsymbol{Q} \in \mathbb{R}^{p \times p} : \boldsymbol{Q}^\top \boldsymbol{Q} = \boldsymbol{I}\}$, defines a well-known group. Thus, the condition that $\mathcal{G}$ is a group of isometries is equivalent to $\mathcal{G}$ being a subgroup of $\mathcal{O}(p)$—i.e., a subset of $\mathcal{O}(p)$ satisfying the group axioms.

Intuitively, by requiring $\mathcal{G}$ to be a group of functions, we ensure that the alignment procedure is symmetric—i.e. it is equivalent to transform $\boldsymbol{X}_i^\phi$ to match $\boldsymbol{X}_j^\phi$, or transform the latter to match the former. Further, by requiring each $\boldsymbol{T} \in \mathcal{G}$ to be an isometry, we ensure that the underlying metric (Euclidean distance for Proposition 1; angular distance for Proposition 2) preserves its key properties.

Together, these propositions define a broad class of metrics as we enumerate below. For simplicity, we assume that $n_i = n_j = n$ in the examples below, with the understanding that a PCA or zero-padding preprocessing step has been performed in the case of dissimilar network sizes. This enables us to express the metrics as functions of the raw activations, i.e. functions $\mathbb{R}^{m \times n} \times \mathbb{R}^{m \times n} \mapsto \mathbb{R}_+$.

**Permutation invariance**   The most stringent notion of representational similarity is to demand that neurons are one-to-one matched across networks. If we set the feature map to be the identity function, i.e., $\boldsymbol{X}_i^\phi = \boldsymbol{X}_i$ for all $i$, then:

$$d_{\mathcal{P}}(\boldsymbol{X}_i, \boldsymbol{X}_j) = \min_{\boldsymbol{\Pi} \in \mathcal{P}(n)} \|\boldsymbol{X}_i - \boldsymbol{X}_j \boldsymbol{\Pi}\| \tag{6}$$

defines a metric by Proposition 1 since the set of permutation matrices, $\mathcal{P}(n)$, is a subgroup of $\mathcal{O}(n)$. To evaluate this metric we must optimize over the set of neuron permutations to align the two networks. This can be reformulated (see Supplement C) as a fundamental problem in combinatorial optimization known as the linear assignment problem [26]. Exploiting an algorithm due to Jonker and Volgenant [27, 28] we can solve this problem in $O(n^3)$ time. The overall runtime for evaluating Eq. (6) is $O(mn^2 + n^3)$, since we must evaluate $\boldsymbol{X}_i^\top \boldsymbol{X}_j$ to formulate the assignment problem.

**Rotation invariance**   Let $\boldsymbol{C} = \boldsymbol{I}_m - (1/m)\boldsymbol{1}\boldsymbol{1}^\top$ denote an $m \times m$ *centering matrix*, and consider the feature mapping $\phi_1$ which mean-centers the columns, $\phi_1(\boldsymbol{X}_i) = \boldsymbol{X}_i^{\phi_1} = \boldsymbol{C}\boldsymbol{X}_i$. Then,

$$d_1(\boldsymbol{X}_i, \boldsymbol{X}_j) = \min_{\boldsymbol{Q} \in \mathcal{O}} \|\boldsymbol{X}_i^{\phi_1} - \boldsymbol{X}_j^{\phi_1} \boldsymbol{Q}\| \tag{7}$$

defines a metric by Proposition 1, and is equivalent to the *Procrustes size-and-shape distance* with reflections [15]. Further, by Proposition 2,

$$\theta_1(\boldsymbol{X}_i, \boldsymbol{X}_j) = \min_{\boldsymbol{Q} \in \mathcal{O}} \ \arccos \frac{\langle \boldsymbol{X}_i^{\phi_1}, \boldsymbol{X}_j^{\phi_1} \boldsymbol{Q}\rangle}{\|\boldsymbol{X}_i^{\phi_1}\|\|\boldsymbol{X}_j^{\phi_1}\|} \tag{8}$$

defines another metric, and is closely related to the Riemannian distance on Kendall's shape space [15]. To evaluate Eqs. (7) and (8), we must optimize over the set of orthogonal matrices to find the best alignment. This also maps onto a fundamental optimization problem known as the *orthogonal Procrustes problem* [29, 30], which can be solved in closed form in $O(n^3)$ time. As in the permutation-invariant metric described above, the overall runtime is $O(mn^2 + n^3)$.

**Linear invariance**   Consider a partial whitening transformation, parameterized by $0 \leq \alpha \leq 1$:

$$\boldsymbol{X}^{\phi_\alpha} = \boldsymbol{C}\boldsymbol{X}(\alpha\boldsymbol{I}_n + (1-\alpha)(\boldsymbol{X}^\top \boldsymbol{C}\boldsymbol{X})^{-1/2}) \tag{9}$$

Note that $\boldsymbol{X}^\top \boldsymbol{C}\boldsymbol{X}$ is the empirical covariance matrix of $\boldsymbol{X}$. Thus, when $\alpha = 0$, Eq. (9) corresponds to ZCA whitening [31], which intuitively removes invertible linear transformations from the representations. Thus, when $\alpha = 0$ the metric outlined below treats $\boldsymbol{X}_i \sim \boldsymbol{X}_j$ if there exists an affine transformation that relates them: $\boldsymbol{X}_i = \boldsymbol{X}_j \boldsymbol{W} + \boldsymbol{b}$ for some $\boldsymbol{W} \in \mathbb{R}^{n \times n}$ and $\boldsymbol{b} \in \mathbb{R}^n$. When $\alpha = 1$, Eq. (9) reduces to the mean-centering feature map used above.

Using orthogonal alignments within this feature space leads to a metric that is related to CCA. First, let $\rho_1 \geq \ldots \geq \rho_n \geq 0$ denote the singular values of $(\boldsymbol{X}_i^{\phi_\alpha})^\top (\boldsymbol{X}_j^{\phi_\alpha})/\|\boldsymbol{X}_i^{\phi_\alpha}\|\|\boldsymbol{X}_j^{\phi_\alpha}\|$. One can show that

$$\theta_\alpha(\boldsymbol{X}_i, \boldsymbol{X}_j) = \min_{\boldsymbol{Q} \in \mathcal{O}} \ \arccos \frac{\langle \boldsymbol{X}_i^{\phi_\alpha}, \boldsymbol{X}_j^{\phi_\alpha} \boldsymbol{Q}\rangle}{\|\boldsymbol{X}_i^{\phi_\alpha}\|\|\boldsymbol{X}_j^{\phi_\alpha}\|} = \arccos(\textstyle\sum_\ell \rho_\ell), \tag{10}$$

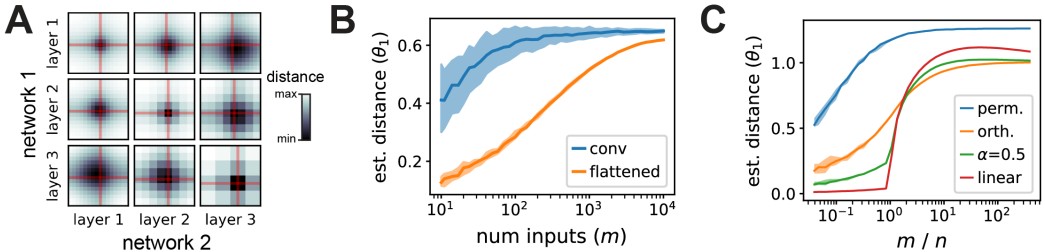

Figure 3: *(A)* Each heatmap shows a brute-force search over the shift parameters along the width and height dimensions of a pair of convolutional layers compared across two networks. The optimal shifts are typically close to zero (red lines). *(B)* Impact of sample size, $m$, on flattened and convolutional metrics with orthogonal invariance. The convolutional metric approaches its final value faster than the flattened metric, which is still increasing even at the full size of the CIFAR-10 test set ($m = 10^4$). *(C)* Impact of sample density, $m/n$, on metrics invariant to permutation, orthogonal, regularized linear ($\alpha = 0.5$), and linear transformations. Shaded regions mark the $10^{th}$ and $90^{th}$ percentiles across shuffled repeats. Further details are provided in Supplement E.

and we can see from Proposition 2 that this defines a metric for any $0 \leq \alpha \leq 1$. When $\alpha = 0$, the values $\rho_1, \ldots, \rho_n$ are proportional to the canonical correlation coefficients, with $1/n$ being the factor of proportionality. When $\alpha > 0$, these values can be viewed as ridge regularized canonical correlation coefficients [32]. See Supplement C for further details. Past works [6, 7] have used the average canonical correlation as a measure of representational similarity. When $\alpha = 0$, the average canonical correlation is given by $\sum_\ell \rho_\ell = \cos \theta_0(\boldsymbol{X}_i, \boldsymbol{X}_j)$. Thus, if we apply $\arccos(\cdot)$ to the average canonical correlation, we modify the calculation to produce a proper metric (see Fig. 4A). Since the covariance is often ill-conditioned or singular in practice, setting $\alpha > 0$ to regularize the calculation is also typically necessary.

**Nonlinear invariances** We discuss feature maps that enable nonlinear notions of equivalence, and which relate to kernel CCA [33] and CKA [9], in Supplement C.

### 2.4 Metrics for convolutional layers

In deep networks for image processing, each convolutional layer produces a $h \times w \times c$ array of activations, whose axes respectively correspond to image height, image width, and channels (number of convolutional filters). If stride-1 circular convolutions are used, then applying a circular shift along either spatial dimension produces the same shift in the layer's output. It is natural to reflect this property, known as translation equivariance [23], in the equivalence relation on layer representations. Supposing that the feature map preserves the shape of the activation tensor, we have $\boldsymbol{X}_k^\phi \in \mathbb{R}^{m \times h \times w \times c}$ for neural networks indexed by $k \in 1, \ldots, K$. Letting $\mathcal{S}(n)$ denote the group of $n$-dimensional circular shifts (a subgroup of the permutation group) and '$\otimes$' denote the Kronecker product, we propose:

$$\boldsymbol{X}_i^\phi \sim \boldsymbol{X}_j^\phi \iff \mathrm{vec}(\boldsymbol{X}_i^\phi) = (\boldsymbol{I} \otimes \boldsymbol{S}_1 \otimes \boldsymbol{S}_2 \otimes \boldsymbol{Q})\mathrm{vec}(\boldsymbol{X}_j^\phi) \qquad (11)$$

for some $\boldsymbol{S}_1 \in \mathcal{S}(h), \boldsymbol{S}_2 \in \mathcal{S}(w), \boldsymbol{Q} \in \mathcal{O}(c)$, as the desired equivalence relation. This relation allows for orthogonal invariance across the channel dimension but only shift invariance across the spatial dimensions. The mixed product property of Kronecker products, $(\boldsymbol{A} \otimes \boldsymbol{B})(\boldsymbol{C} \otimes \boldsymbol{D}) = \boldsymbol{AB} \otimes \boldsymbol{CD}$, ensures that the overall transformation maintains the group structure and remains an isometry. Figure 2B uses a toy dataset (stacked MNIST digits) to show that this metric is sensitive to differences in spatial activation patterns, but insensitive to coherent spatial translations across channels. In contrast, metrics that ignore the convolutional structure (as in past work [6, 9]) treat very different spatial patterns as identical representations.

Evaluating Eq. (11) requires optimizing over spatial shifts in conjuction with solving a Procrustes alignment. If we fit the shifts by an exhaustive brute-force search, the overall runtime is $O(mh^2w^2c^2 + hwc^3)$, which is costly if this calculation is repeated across a large collection of networks. In practice, we observe that the optimal shift parameters are typically close to zero (Fig. 3A). This motivates the more stringent equivalence relation:

$$\boldsymbol{X}_i^\phi \sim \boldsymbol{X}_j^\phi \iff \mathrm{vec}(\boldsymbol{X}_i^\phi) = (\boldsymbol{I} \otimes \boldsymbol{I} \otimes \boldsymbol{I} \otimes \boldsymbol{Q})\mathrm{vec}(\boldsymbol{X}_j^\phi) \quad \text{for some } \boldsymbol{Q} \in \mathcal{Q}, \qquad (12)$$

which has a more manageable runtime of $O(mhwc^2 + c^3)$. To evaluate the metrics implied by Eq. (12), we can simply reshape each $\boldsymbol{X}_k^\phi$ from a $(m \times h \times w \times c)$ tensor into a $(mhw \times c)$ matrix and apply the Procrustes alignment procedure as done above for previous metrics. In contrast, the "flattened metric" in Fig. 2B reshapes the features into a $(m \times hwc)$ matrix, resulting in a more computationally expensive alignment that runs in $O(mh^2w^2c^2 + h^3w^3c^3)$ time.

## 2.5   How large of a sample size is needed?

An important issue, particularly in neurobiological applications, is to determine the number of network inputs, $m$, and neurons, $n$, that one needs to accurately infer the distance between two network representations [12]. Reasoning about these questions rigorously requires a probabilistic perspective of neural representational similarity, which is missing from current literature and which we outline in Supplement D for generalized shape metrics. Intuitively, looser equivalence relations are achieved by having more flexible alignment operations (e.g. nonlinear instead of linear alignments). Thus, looser equivalence relations require more sampled inputs to prevent overfitting. Figure 3B-C show that this intuition holds in practice for data from deep convolutional networks. Metrics with looser equivalence relations—the "flattened" metric in panel B, or e.g. the linear metric in panel C—converge slower to a stable estimate as $m$ is increased.

## 2.6   Modeling approaches and conceptual insights

Generalized shape metrics facilitate several new modeling approaches and conceptual perspectives. For example, a collection of representations from $K$ neural networks can, in certain cases, be interpreted and visualized as $K$ points on a smooth manifold (see Fig. 1). This holds rigorously due to the *quotient manifold theorem* [34] so long as $\mathcal{G}$ is not a finite set (e.g. corresponding to permutation) and all matrices are full rank in the feature space. This geometric intuition can be made even stronger when $\mathcal{G}$ corresponds to a connected manifold, such as $\mathcal{SO}(p)$. In this case, it can be shown that the geodesic distance between two neural representations coincides with the metrics we defined in Propositions 1 and 2 (see Supplement C, and [15]). This result extends the well-documented manifold structure of *Kendall's shape space* [35].

Viewing neural representations as points on a manifold is not a purely theoretical exercise—several models can be adapted to manifold-valued data (e.g. principal geodesic analysis [36] provides a generalization of PCA), and additional adaptions are an area of active research [37]. However, there is generally no simple connection between these curved geometries and the flat geometries of Euclidean or Hilbert spaces [38].[1] Unfortunately, the majority of off-the-shelf machine learning tools are incompatible with the former and require the latter. Thus, we can resort to a heuristic approach: the set of $K$ representations can be embedded into a Euclidean space that approximately preserves the pairwise shape distances. One possibility, employed widely in shape analysis, is to embed points in the tangent space of the manifold at a reference point [41, 42]. Another approach, which we demonstrate below with favorable results, is to optimize the vector embedding directly via multi-dimensional scaling [43, 44].

## 3   Applications and Results

We analyzed two large-scale public datasets spanning neuroscience (Allen Brain Observatory, ABO; Neuropixels - visual coding experiment; [21]) and deep learning (NAS-Bench-101; [22]). We constructed the ABO dataset by pooling recorded neurons from $K = 48$ anatomically defined brain regions across all sessions; each $\boldsymbol{X}_k \in \mathbb{R}^{m \times n}$ was a dimensionally reduced matrix holding the neural responses (summarized by $n = 100$ principal components) to $m = 1600$ movie frames (120 second clip, "natural movie three"). The full NAS-Bench-101 dataset contains 423,624 architectures; however, we analyze a subset of $K = 2000$ networks for simplicity. In this application each $\boldsymbol{X}_k \in \mathbb{R}^{m \times n}$ is a representation from a specific network layer, with $(m, n) \in \{(32^2 \times 10^5, 128), (16^2 \times 10^5, 256), (8^2 \times 10^5, 512), (10^5, 512)\}$. Here, $n$ corresponds to the number of channels and $m$ is the product of the number of test set images ($10^5$) and the height and width dimensions of the convolutional layer—i.e., we use equivalence relation in Eq. (12) to evaluate dissimilarity.

---

[1]However, see [39] for a conjectured relationship and [40] for a result in the special case of 2D shapes.

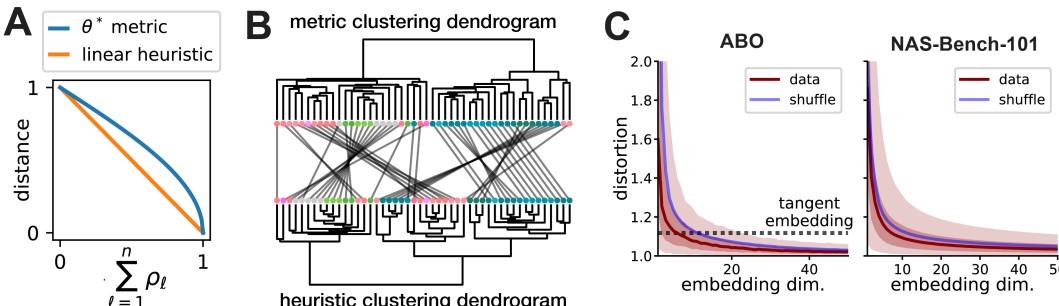

Figure 4: (A) Comparison of metric and linear heuristic. (B) Metric and linear heuristic produce discordant hierarchical clusterings of brain areas in the ABO dataset. Leaves represent brain areas that are clustered by representational similarity (see Fig. 1C), colored by Allen reference atlas, and ordered to maximize dendrogram similarities of adjacent leaves. In the middle, grey lines connect leaves corresponding to the same brain region across the two dendrograms. (C) ABO and NAS-Bench-101 datasets can be accurately embedded into Euclidean spaces. Dark red line shows median distortion. Light red shaded region corresponds to 5th to 95th percentiles of distortion, dark red shaded corresponds to interquartile range. The mean distortion of a null distribution over representations (blue line) was generated by shuffling the $m$ inputs independently in each network.

**Triangle inequality violations can occur in practice when using existing methods.** As mentioned above, a dissimilarity measure based on the mean canonical correlation, $1 - \sum_\ell \rho_\ell$, has been used in past work [7, 10]. We refer to this as the "linear heuristic." A slight reformulation of this calculation, $\arccos\left(\sum_\ell \rho_\ell\right)$, produces a metric that satisfies the triangle inequality (see Eq. (10)). Figure 4A compares these calculations as a function of the average (regularized) canonical correlation: one can see that $\arccos(\cdot)$ is approximately linear when the mean correlation is near zero, but highly nonlinear when the mean correlation is near one. Thus, we reasoned that triangle inequality violations are more likely to occur when $K$ is large and when many network representations are close to each other. Both ABO and NAS-Bench-101 datasets satisfy these conditions, and in both cases we observed triangle inequality violations by the linear heuristic with full regularization ($\alpha = 1$): 17/1128 network pairs in the ABO dataset had at least one triangle inequality violation, while 10128/100000 randomly sampled network pairs contained violations in the NAS-Bench-101 Stem layer dataset. We also examined a standard version of RSA that quantifies similarity via Spearman's rank correlation coefficient [8]. Similar to the results above, we observed violations in 14/1128 pairs of networks in the ABO dataset.

Overall, these results suggest that generalized shape metrics correct for triangle inequality violations that do occur in practice. Depending on the dataset, these violations may be rare (~1% occurrence in ABO) or relatively common (~10% in the Stem layer of NAS-Bench-101). These differences can produce quantitative discrepancies in downstream analyses. For example, the dendrograms produced by hierarchical clustering differ depending on whether one uses the linear heuristic or the shape distance (~85.1% dendrogram similarity as quantified by the method in [45]; see Fig. 4B).

**Neural representation metric spaces can be approximated by Euclidean spaces.** Having established that neural representations can be viewed as elements in a metric space, it is natural to ask if this metric space is, loosely speaking, "close to" a Euclidean space. We used standard multidimensional scaling methods (SMACOF, [43]; implementation in [46]) to obtain a set of embedded vectors, $\boldsymbol{y}_i \in \mathbb{R}^L$, for which $\theta_1(\boldsymbol{X}_i^\phi, \boldsymbol{X}_j^\phi) \approx \|\boldsymbol{y}_i - \boldsymbol{y}_j\|$ for $i, j \in 1, \dots, K$. The embedding dimension $L$ is a user-defined hyperparameter. This problem admits multiple formulations and optimization strategies [44], which could be systematically explored in future work. Our simple approach already yields promising results: we find that moderate embedding dimensions ($L \approx 20$) is sufficient to produce high-quality embeddings. We quantify the embedding distortions multiplicatively [47]:

$$\max\left(\theta_1(\boldsymbol{X}_i^\phi, \boldsymbol{X}_j^\phi)/\|\boldsymbol{y}_i - \boldsymbol{y}_j\|; \ \|\boldsymbol{y}_i - \boldsymbol{y}_j\|/\theta_1(\boldsymbol{X}_i^\phi, \boldsymbol{X}_j^\phi)\right) \tag{13}$$

for each pair of networks $i, j \in 1, \dots K$. Plotting the distortions as a function of $L$ (Fig. 4C), we see that they rapidly decrease, such that 95% of pairwise distances are distorted by, at most, ~5% (ABO data) or 10% (NAS-Bench-101) for sufficiently large $L$. Past work [10] has used multidimensional scaling heuristically to visualize collections of network representations in $L = 2$ dimensions. Our results here suggest that such a small value of $L$, while being amenable to visualization, results in a highly distorted embedding. It is noteworthy that the situation improves dramatically when

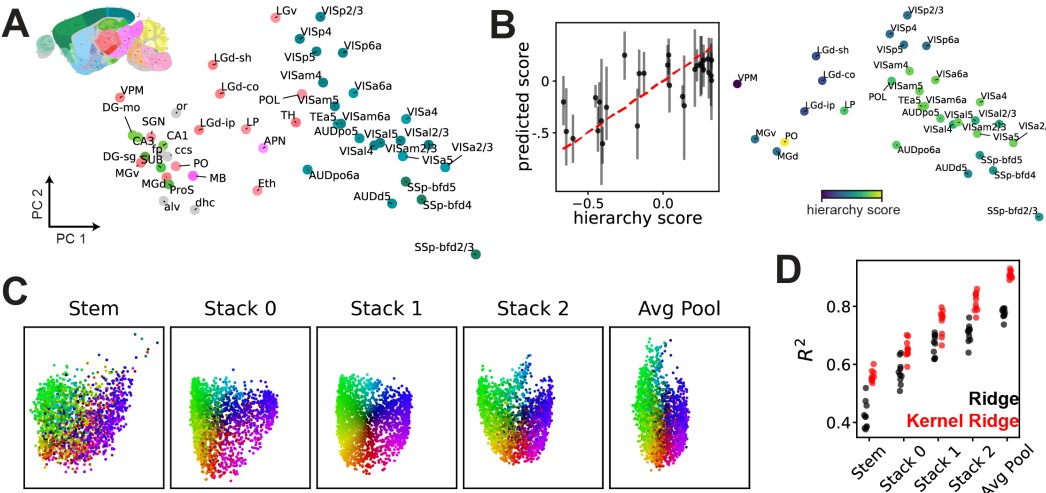

Figure 5: (A) PCA visualization of representations across 48 brain regions in the ABO dataset. Areas are colored by the reference atlas (see inset), illustrating a functional clustering of regions that maps onto anatomy. (B) *Left*, kernel regression predicts anatomical hierarchy [48] from embedded representations (see Supplement E). *Right*, PCA visualization of 31 areas labeled with hierarchy scores. (C) PCA visualization of 2000 network representations (a subset of NAS-Bench-101) across five layers, showing global structure is preserved across layers. Each network is colored by its position in the "Stack 1" layer (the middle of the architecture). (D) Embeddings of NAS-Bench-101 representations are predictive of test set accuracy, *even in very early layers*.

$L$ is even modestly increased. While we cannot easily visualize these higher-dimensional vector embeddings, we can use them as features for downstream modeling tasks. This is well-motivated as an approximation to performing model inference in the true metric space that characterizes neural representations [47].

**Anatomical structure and hierarchy is reflected in ABO representations.** We can now collect the $L$-dimensional vector embeddings of $K$ network representations into a matrix $\boldsymbol{Z} \in \mathbb{R}^{K \times L}$. The results in Fig. 4C imply that the distance between any two rows, $\|\boldsymbol{z}_i - \boldsymbol{z}_j\|$, closely reflects the distance between network representations $i$ and $j$ in shape space. We applied PCA to $\boldsymbol{Z}$ to visualize the $K = 48$ brain regions and found that anatomically related brain regions indeed were closer together in the embedded space (Fig. 5A): cortical and sub-cortical regions are separated along PC 1, and different layers of the same region (e.g. layers 2/3, 4, 5, and 6a of VISp) are clustered together. As expected from Fig. 4C, performing multidimensional scaling directly to a low-dimensional space ($L = 2$, as done in [10]) results in a qualitatively different outcome with distorted geometry (see Supplement E). Additionally, we used $\boldsymbol{Z}$ to fit an ensembled kernel regressor to predict an anatomical hierarchy score (defined in [48]) from the embedded vectors (Fig. 5B). Overall, these results demonstrate that the geometry of the learned embedding is scientifically interpretable and can be exploited for novel analyses, such as nonlinear regression. To our knowledge, the fine scale anatomical parcellation used here is novel in the context of representational similarity studies.

**NAS-Bench-101 representations show persistent structure across layers.** Since we collected representations across five layers in each deep network, the embedded representation vectors form a set of five $K \times L$ matrices, $\{\boldsymbol{Z}_1, \boldsymbol{Z}_2, \boldsymbol{Z}_3, \boldsymbol{Z}_4, \boldsymbol{Z}_5\}$. We aligned these embeddings by rotations in $\mathbb{R}^L$ via Procrustes analysis, and then performed PCA to visualize the $K = 2000$ network representations from each layer in a common low-dimensional space. We observe that many features of the global structure are remarkably well-preserved—two networks that are close together in the `Stack1` layer are assigned similar colors in Fig. 5C, and are likely to be close together in the other four layers. This preservation of representational similarity across layers suggests that even early layers contain signatures of network performance, which we expect to be present in the `AvgPool` layer. Indeed, when we fit ridge and RBF kernel ridge regressors to predict test set accuracy from representation embeddings, we see that even early layers support moderately good predictions (Fig. 5D). This is particularly surprising for the `Stem` layer. This is the first layer in each network, and its architecture is identical for all networks. Thus, the differences that are detected in the `Stem` layer result only from

differences in backpropagated gradients. Again, these results demonstrate the ability of generalized shape metrics to incorporate neural representations into analyses with greater scale ($K$ corresponding to thousands of networks) and complexity (nonlinear kernel regression) than has been previously explored.

## 4    Conclusion and Limitations

We demonstrated how to ground analyses of neural representations in proper metric spaces. By doing so, we capture a number of theoretical advantages [16–20]. Further, we suggest new practical modeling approaches, such as using Euclidean embeddings to approximate the representational metric spaces. An important limitation of our work, as well as the past works we build upon, is the possibility that representational geometry is only loosely tied to higher-level algorithmic principles of network function [10]. On the other hand, analyses of representational geometry may provide insight into lower-level implementational principles [49]. Further, these analyses are highly scalable, as we demonstrated by analyzing thousands of networks—a much larger scale than is typically considered.

We used simple metrics (extensions of regularized CCA) in these analyses, but metrics that account for nonlinear transformations across neural representations are also possible as we document in Supplement C. The utility of these nonlinear extensions remains under-investigated and it is possible that currently popular linear methods are insufficient to capture structures of interest. For example, the topology of neural representations has received substantial interest in recent years [50–53]. Generalized shape metrics do not directly capture these topological features, and future work could consider developing new metrics that do so. A variety of recent developments in topological data analysis may be useful towards this end [54–56].

Finally, several of the metrics we described can be viewed as geodesic distances on Riemannian manifolds [35]. Future work would ideally exploit methods that are rigorously adapted to such manifolds, which are being actively developed [37]. Nonetheless, we found that optimized Euclidean embeddings, while only approximate, provide a practical off-the-shelf solution for large-scale surveys of neural representations.

**Acknowledgments**

We thank Ian Dryden (Florida International University), Søren Hauberg (Technical University of Denmark), and Nina Miolane (UC Santa Barbara) for fruitful discussions. A.H.W. was supported by the National Institutes of Health BRAIN initiative (1F32MH122998-01), and the Wu Tsai Stanford Neurosciences Institute Interdisciplinary Scholar Program. E. K. was supported by the Wu Tsai Stanford Neurosciences Institute Interdisciplinary Graduate Fellows Program. S.W.L. was supported by grants from the Simons Collaboration on the Global Brain (SCGB 697092) and the NIH BRAIN Initiative (U19NS113201 and R01NS113119).

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
