# Supplementary Information: Generalized Shape Metrics on Neural Representations

This supplement is organized into five sections. First, in Supplement A, we review background material on metric spaces and other relevant mathematical concepts. In Supplement B, we prove the two propositions that appear in the main text. Supplement C collects together several miscellaneous results which demonstrate that generalized shape metrics include similarity measures based on CCA, kernel CCA, and geodesic distance on Kendall's shape space. In Supplement D, we outline an extension and reinterpretation of generalized shape metrics to stochastic random variables. This extension represents a rich opportunity for future research and also provides a better foundation to interpret the results presented in Fig. 3 of the main text, which empirically characterize the number of images needed to estimate the distance between two neural networks. Finally, in Supplement E, we collect additional methodological details about the experiments we present in the main text.

## A    Background

### A.1    Notation

Vectors in real coordinate space are denoted in boldface with lowercase letters, e.g. $\boldsymbol{x} \in \mathbb{R}^n$. Matrices are denoted in boldface with uppercase letters, e.g. $\boldsymbol{X} \in \mathbb{R}^{m \times n}$. We use the same notation to denote linear operators, e.g. $\boldsymbol{T} \in \mathcal{G}$ where $\mathcal{G}$ is a set of linear operators.

Letters in regular type face, e.g. $x$ or $X$, may denote scalars or elements of some abstract vector space, with the distinction being made clear from context. For example, the space of random variables with outcomes over $\mathbb{R}^n$ defines a vector space that we will see is compatible with the basic framework of generalized shape metrics. This extension of shape metrics to stochastic layers and neural responses is outlined in Supplement D.

If $\boldsymbol{T}$ is a linear operator on some vector space, and $X$ is a vector within this space, we will use $\boldsymbol{T}X$ to denote the transformation $X \mapsto \boldsymbol{T}(X)$. Further, if $\boldsymbol{T}_1$ and $\boldsymbol{T}_2$ are linear operators, we write $\boldsymbol{T}_1\boldsymbol{T}_2 X$ in place of $\boldsymbol{T}_1(\boldsymbol{T}_2(X))$, and we use $\boldsymbol{T}_1\boldsymbol{T}_2$ to denote the composition of the two linear operators. These notational choices intuitively draw parallels with matrix-vector and matrix-matrix multiplication, respectively.

### A.2    Metrics

Here we revisit our definition of a metric given in the main text to provide more rigorous details and clarify the role of the equivalence relation.

**Definition 1.** *A **metric** on a set $\mathcal{S}$ is a function $\mathcal{S} \times \mathcal{S} \mapsto \mathbb{R}_+$, which satisfies, for all $X, Y, M \in \mathcal{S}$, the following three conditions:*

- *Identity.* $d(X, Y) = 0$ *if and only if* $X = Y$

- *Symmetry.* $d(X, Y) = d(Y, X)$

- *Triangle Inequality.* $d(X, Y) \leq d(X, M) + d(M, Y)$

We have seen that it is useful to relax the first condition (*Identity*) to an equivalence relation. That is, rather than strict equality, we demand that $d(X, Y) = 0$ if and only if $X \sim Y$, for some specified equivalence relation $\sim$. In this scenario, the distance function is *not*, strictly speaking, a metric on $\mathcal{S}$. However, it still does define a metric on the appropriate *quotient set*, which we now define.

**Definition 2.** *Let $\sim$ denote an equivalence relation defined on some set $\mathcal{S}$. Then given any $M \in \mathcal{S}$, we can define the set of all elements equivalent to $M$ as $\{X \in \mathcal{S} \mid X \sim M\}$, which is called the **equivalence class** of the element $M$. The set of all equivalence classes, denoted $\mathcal{S}/\sim$, is called the **quotient set** of $\mathcal{S}$ with respect to the specified equivalence relation.*

For example, the Euclidean distance $\|\boldsymbol{x} - \boldsymbol{y}\|$ is a metric on the set of vectors in $\mathbb{R}^n$. The angular distance $\arccos(\boldsymbol{x}^\top \boldsymbol{y} / \sqrt{\boldsymbol{x}^\top \boldsymbol{x} \cdot \boldsymbol{y}^\top \boldsymbol{y}})$ is not a metric on $\mathbb{R}^n$, but it defines a metric between sets of points contained in rays emanating from the origin (i.e. points in $\mathbb{R}^n$ with an equivalence relation given by nonnegative scaling). These technical distinctions above are not central to our story, so we will often refer to a function as a "metric" without explicitly defining what set it acts upon. In all cases, it should be understood as the quotient set defined by the specified equivalence relation.

## A.3  Hilbert spaces

A **vector space** $\mathcal{H}$ is a collection of objects (called vectors) that are equipped with two operations: vector addition (given $X \in \mathcal{H}$ and $Y \in \mathcal{H}$ we have $X + Y \in \mathcal{H}$) and scalar multiplication (given $X \in \mathcal{H}$ and $\alpha \in \mathbb{R}$ we have $\alpha X \in \mathcal{H}$). An **inner product space** is a vector space that is additionally equipped with a function $\mathcal{H} \times \mathcal{H} \mapsto \mathbb{R}$, called the *inner product*, which is denoted with angle brackets $\langle \cdot, \cdot \rangle$ and satisfies:

- *Symmetry.* $\langle X, Y \rangle = \langle Y, X \rangle$

- *Linearity.* $\langle Z + \alpha X, Y \rangle = \langle Z, Y \rangle + \alpha \langle X, Y \rangle$

- *Positive Definiteness.* $\langle X, X \rangle \geq 0$ with equality if and only if $X = 0$

A **Hilbert space** is an inner product space that satisfies an additional technical requirement (all Cauchy sequences of vectors in $\mathcal{H}$ converge to a limit in $\mathcal{H}$).

The set of vectors in $\mathbb{R}^n$ defines a Hilbert space, where the inner product corresponds to the usual dot product. Similarly, the set of matrices in $\mathbb{R}^{m \times n}$, equipped with the Frobenius inner product $\langle \boldsymbol{X}, \boldsymbol{Y} \rangle = \text{Tr}[\boldsymbol{X}^\top \boldsymbol{Y}]$ also defines a Hilbert space. In Supplement D, we will exploit the fact that random vectors over $\mathbb{R}^n$ also define a Hilbert space where the inner product is given by the expectation of the dot product. This enables us to extend the framework of generalized shape metrics to stochastic neural layers.

## A.4  Euclidean and Angular Distances in Hilbert Spaces

One of the most fundamental properties of a Hilbert space is the **Cauchy-Schwarz inequality**,

$$|\langle X, Y \rangle| \leq \|X\| \|Y\| \quad \text{for all} \quad (X, Y) \in \mathcal{H} \times \mathcal{H}, \tag{1}$$

which can be derived from the properties of the inner product. Using this, we can verify that the norm is sub-additive:

$$\|X + Y\| \leq \|X\| + \|Y\| \quad \text{for all} \quad (X, Y) \in \mathcal{H} \times \mathcal{H}, \tag{2}$$

Defining $d_{\text{euc}}(X, Y) = \|X - Y\|$ to be the generalization of Euclidean distance to Hilbert spaces, we see that triangle inequality follows immediately:

$$d_{\text{euc}}(X, Y) = \|X - Y\| = \|X - M + M - Y\| \leq \|X - M\| + \|M - Y\| = d_{\text{euc}}(X, M) + d_{\text{euc}}(M, Y) \tag{3}$$

for all choices of $X$, $Y$, and $M$ in $\mathcal{H}$. Euclidean distance evidently satisfies the remaining two properties of a metric—symmetry and nonnegativity.

The angular distance is defined as:

$$d_\theta(X, Y) = \arccos\left[\frac{\langle X, Y \rangle}{\|X\|\|Y\|}\right] \tag{4}$$

The Cauchy-Schwarz inequality implies that the argument to $\arccos(\cdot)$ is always within its domain (i.e. on the interval $[-1, 1]$). The angular distance is a metric over equivalence classes defined by nonnegative scaling: formally, $X \sim Y$ if and only if there exists an $s > 0$ such that $X = sY$. Geometrically, one can think of $d_\theta(X, Y)$ as the geodesic path length between points on a sphere. Intuitively, this is nonnegative, symmetric, and obeys the triangle inequality. We provide a short proof that the triangle inequality is indeed satisfied below.

***Proof:*** *Angular distance satisfies the triangle inequality.* Consider three unit-norm vectors: $X$, $Y$, and $Z$. The triangle inequality trivially holds if any pair of $X$, $Y$, and $Z$ are equal, so we can assume $X$, $Y$, and $Z$ are distinct. Now define two vectors $U$ and $V$ as follows:

$$U = X - Y\langle X, Y \rangle \tag{5}$$
$$V = Z - Y\langle Z, Y \rangle \tag{6}$$

Note that $\langle U, Y \rangle = 0$ and $\langle V, Y \rangle = 0$. Thus, we can interpret $U$ as the part of $X$ that is orthogonal to $Y$. Likewise, we can interpret $V$ as the part $Z$ that is orthogonal to $Y$. Further, we have:

$$X = Y\langle X, Y \rangle + U\langle X, U \rangle = Y\cos\theta_{XY} + U\sin\theta_{XY} \tag{7}$$
$$Z = Y\langle Z, Y \rangle + V\langle Z, V \rangle = Y\cos\theta_{ZY} + V\sin\theta_{ZY} \tag{8}$$

where we introduced the shorthand $\theta_{XY} = d_\theta(X, Y)$ for concision. Now,

$$\cos\theta_{XZ} = \langle X, Z \rangle = \langle Y\cos\theta_{XY} + U\sin\theta_{XY}, Y\cos\theta_{ZY} + V\sin\theta_{ZY} \rangle \tag{9}$$
$$= \cos\theta_{XY}\cos\theta_{ZY} + \langle U, V \rangle\sin\theta_{XY}\sin\theta_{ZY} \tag{10}$$
$$\geq \cos\theta_{XY}\cos\theta_{ZY} - \sin\theta_{XY}\sin\theta_{ZY} \tag{11}$$
$$= \cos(\theta_{XY} + \theta_{ZY}) \tag{12}$$

On line (10), many terms simplify since $\langle Y, Y \rangle = 1$, and $\langle U, Y \rangle = \langle V, Y \rangle = 0$. To introduce the inequality on line (11), notice that the Cauchy-Schwarz inequality implies $\langle U, V \rangle \geq -1$. Thus, replacing $\langle U, V \rangle$ with $-1$ produces a lower bound on $\cos\theta_{XZ}$ since $\sin\theta_{XY}\sin\theta_{ZY} \geq 0$. The final step on line (12) applies an

elementary identity from trigonometry. Overall, we have $\cos\theta_{XZ} \geq \cos(\theta_{XY} + \theta_{ZY})$. This directly implies the desired triangle inequality, $\theta_{XZ} \leq \theta_{XY} + \theta_{ZY}$, since $\arccos(\cdot)$ is a monotonically decreasing function. $\square$

## A.5 The Orthogonal Group

Another important feature of Hilbert spaces is the notion of an orthogonal transformation. These are linear transformations which preserve the inner product. Below, we also define the familiar transpose operator for a general Hilbert space.

**Definition 3.** *An **orthogonal transformation** on a Hilbert space $\mathcal{H}$ is any linear transformation $\boldsymbol{Q}$, which satisfies $\langle \boldsymbol{Q}X, \boldsymbol{Q}Y \rangle = \langle X, Y \rangle$ for any choice of $X \in \mathcal{H}$ and $Y \in \mathcal{H}$.*

**Definition 4.** *Let $\boldsymbol{W} : \mathcal{V} \mapsto \mathcal{V}$ be a linear transformation on a Hilbert space $\mathcal{V}$. For any choice of $\boldsymbol{W}$, there is a unique linear transformation $\boldsymbol{W}^\top$, called the **transpose** (or adjoint) of $\boldsymbol{W}$, which is denoted $\boldsymbol{W}^\top$ and which satisfies $\langle \boldsymbol{W}X, Y \rangle = \langle X, \boldsymbol{W}^\top Y \rangle$ for any choice of $X \in \mathcal{V}$ and $Y \in \mathcal{V}$.*

Let $\boldsymbol{Q}$ be orthogonal. Since $\langle X, Y \rangle = \langle \boldsymbol{Q}X, \boldsymbol{Q}Y \rangle = \langle X, \boldsymbol{Q}^\top \boldsymbol{Q}Y \rangle$, we see that $\boldsymbol{Q}^\top \boldsymbol{Q}$ is the identity transformation and thus $\boldsymbol{Q}^\top$ and $\boldsymbol{Q}$ are inverses. One can show that these inverses commute, and thus $\boldsymbol{Q}^\top$ is also orthogonal since $\langle \boldsymbol{Q}^\top X, \boldsymbol{Q}^\top Y \rangle = \langle \boldsymbol{Q}\boldsymbol{Q}^\top X, \boldsymbol{Q}\boldsymbol{Q}^\top Y \rangle = \langle X, Y \rangle$. Finally, let $\boldsymbol{Q}_1$ and $\boldsymbol{Q}_2$ be any pair of orthogonal transformations on $\mathcal{V}$. Then, the composition of these transformations $\boldsymbol{Q}_2 \boldsymbol{Q}_1$ is evidently orthogonal, since: $\langle X, Y \rangle = \langle \boldsymbol{Q}_1 X, \boldsymbol{Q}_1 Y \rangle = \langle \boldsymbol{Q}_2 \boldsymbol{Q}_1 X, \boldsymbol{Q}_2 \boldsymbol{Q}_1 Y \rangle$.

In summary, we have just shown that the inverse of every orthogonal matrix is also orthogonal and orthogonal transformations are closed under composition. This shows that the set of orthogonal transformations on a Hilbert space fulfills the axioms of a **group**, as defined below:

**Definition 5.** *A **group** is a set $\mathcal{G}$ equipped with a binary operation that maps two elements of $\mathcal{G}$ onto another element of $\mathcal{G}$, which satisfies:*

1. *Associativity: For all $\boldsymbol{T}_1, \boldsymbol{T}_2, \boldsymbol{T}_3$ in $\mathcal{G}$, one has $(\boldsymbol{T}_1 \boldsymbol{T}_2)\boldsymbol{T}_3 = \boldsymbol{T}_1(\boldsymbol{T}_2 \boldsymbol{T}_3)$.*

2. *Identity element: There exists a unique element $\boldsymbol{I} \in \mathcal{G}$ such that $\boldsymbol{I}\boldsymbol{T} = \boldsymbol{T}\boldsymbol{I} = \boldsymbol{T}$ for all $\boldsymbol{T} \in \mathcal{G}$.*

3. *Invertibility: For every $\boldsymbol{T} \in \mathcal{G}$ there exists another element $\boldsymbol{T}^{-1} \in \mathcal{G}$ such that $\boldsymbol{T}\boldsymbol{T}^{-1} = \boldsymbol{T}^{-1}\boldsymbol{T} = \boldsymbol{I}$.*

Here, we are only interested in groups of *linear functions*, so the "binary operation" referred to above is function composition (see section A.1 for notational conventions regarding linear operators).

We are particularly interested in groups of (linear) transformations that preserve distances. Such transformations are called (linear) **isometries**, which we define below.

**Definition 6.** *Let $\mathcal{S}$ be a set and let $d : \mathcal{S} \times \mathcal{S} \mapsto \mathbb{R}_+$ be a metric on this set. Then a transformation $\boldsymbol{T} : \mathcal{S} \mapsto \mathcal{S}$, is called an **isometry** on the metric space $(d, \mathcal{S})$ if $d(\boldsymbol{T}X, \boldsymbol{T}Y) = d(X, Y)$ for all $X, Y \in \mathcal{S}$.*

It is easy to see that the orthogonal group is a group of isometries with respect to the (generalized) Euclidean and angular distance metrics. For the Euclidean distance we have:

$$\begin{aligned}
d_{\mathsf{euc}}^2(X, Y) = \|X - Y\|^2 &= \langle X, X \rangle + \langle Y, Y \rangle - 2\langle X, Y \rangle \\
&= \langle \boldsymbol{Q}X, \boldsymbol{Q}X \rangle + \langle \boldsymbol{Q}Y, \boldsymbol{Q}Y \rangle - 2\langle \boldsymbol{Q}X, \boldsymbol{Q}Y \rangle \\
&= \|\boldsymbol{Q}X - \boldsymbol{Q}Y\|^2 = d_{\mathsf{euc}}^2(\boldsymbol{Q}X, \boldsymbol{Q}Y)
\end{aligned} \tag{13}$$

For the angular distance we have:

$$\cos\left[d_\theta(X, Y)\right] = \frac{\langle X, Y \rangle}{\|X\|\|Y\|} = \frac{\langle \boldsymbol{Q}X, \boldsymbol{Q}Y \rangle}{\|\boldsymbol{Q}X\|\|\boldsymbol{Q}Y\|} = \cos\left[d_\theta(\boldsymbol{Q}X, \boldsymbol{Q}Y)\right] \tag{14}$$

# B    Proof of Propositions 1 & 2

Both propositions in the main text follow immediately as special cases of the following result, which states that minimizing any metric over a group of isometries results in a metric on the corresponding quotient space. After proving this result we conclude this section by briefly outlining these special cases.

**Proposition** (A generalization of Propositions 1 & 2). *Let $(g, \mathcal{H})$ be a metric space, where $g : \mathcal{H} \times \mathcal{H} \mapsto \mathbb{R}_+$ denotes the distance function. Let $\mathcal{G}$ be a group of isometries on this metric space. Then the function:*

$$h(X, Y) = \min_{\boldsymbol{T} \in \mathcal{G}} g(X, \boldsymbol{T}Y) \tag{15}$$

defines a metric over the quotient space $\mathcal{H}/\sim$ where the equivalence relation is $X \sim Y$ if and only if $X = \boldsymbol{T}Y$ for some $\boldsymbol{T} \in \mathcal{G}$.

*Proof.* First, define $\boldsymbol{T}_{XY} = \operatorname{argmin}_{\boldsymbol{T} \in \mathcal{G}} g(X, \boldsymbol{T}Y)$. So, $h(X, Y) = g(X, \boldsymbol{T}_{XY}Y)$. Since $g$ is a metric, $g(X, Y) = 0$ if and only if $X = Y$. Thus, $h(X, Y) = 0$ if and only if $X = \boldsymbol{T}_{XY}Y$, or equivalently if $X \sim Y$ by the stated equivalence relation.

Next, we prove that $h(X, Y) = h(Y, X)$. By the group axioms, every element in $\mathcal{G}$ is invertible by another element in the set, so $\boldsymbol{T}_{XY}^{-1} \in \mathcal{G}$. Further, every element of $\mathcal{G}$ is an isometry with respect to $g$. Thus,

$$h(X, Y) = g(X, \boldsymbol{T}_{XY}Y) = g(\boldsymbol{T}_{XY}^{-1}X, \boldsymbol{T}_{XY}^{-1}\boldsymbol{T}_{XY}Y) = g(Y, \boldsymbol{T}_{XY}^{-1}X) \geq g(Y, \boldsymbol{T}_{YX}X) = h(Y, X), \tag{16}$$

where the inequality follows from replacing $\boldsymbol{T}_{XY}^{-1}$ with the optimal $\boldsymbol{T}_{YX} = \operatorname{argmin}_{\boldsymbol{T} \in \mathcal{G}} g(Y, \boldsymbol{T}X)$. However, by the same chain of logic, we also have:

$$h(Y, X) = g(Y, \boldsymbol{T}_{YX}X) = g(\boldsymbol{T}_{YX}^{-1}Y, \boldsymbol{T}_{YX}^{-1}\boldsymbol{T}_{YX}X) = g(X, \boldsymbol{T}_{YX}^{-1}Y) \geq g(X, \boldsymbol{T}_{XY}Y) = h(X, Y). \tag{17}$$

Thus, we have $h(X, Y) \geq h(Y, X)$, but also $h(Y, X) \geq h(X, Y)$. We conclude $h(X, Y) = h(Y, X)$ and $\boldsymbol{T}_{XY}^{-1} = \boldsymbol{T}_{YX}$.

It remains to prove the triangle inequality. This is done by the following sequence:

$$h(X, Y) = g(X, \boldsymbol{T}_{XY}Y) \tag{18}$$
$$\leq g(X, \boldsymbol{T}_{XZ}\boldsymbol{T}_{ZY}Y) \tag{19}$$
$$\leq g(X, \boldsymbol{T}_{XZ}Z) + g(\boldsymbol{T}_{XZ}Z, \boldsymbol{T}_{XZ}\boldsymbol{T}_{ZY}Y) \tag{20}$$
$$= g(X, \boldsymbol{T}_{XZ}Z) + g(Z, \boldsymbol{T}_{ZY}Y) \tag{21}$$
$$= h(X, Z) + h(Z, Y) \tag{22}$$

The first inequality follows from replacing the optimal alignment, $\boldsymbol{T}_{XY}$, with a sub-optimal alignment $\boldsymbol{T}_{XZ}\boldsymbol{T}_{ZY}$. The second inequality follows from the triangle inequality on $g$, after choosing $\boldsymbol{T}_{XZ}Z$ as the midpoint. The penultimate step follows from $\boldsymbol{T}_{XZ}$ being an isometry on $g$. $\square$

**Relation to Proposition 1** The space $\mathcal{H}$ corresponds to $\mathbb{R}^{m \times p}$, which is equipped with the typical Frobenius inner product. The distance function $g$ is Euclidean distance, see eq. (3). The group $\mathcal{G}$ corresponds to any group of linear isometries which can be expressed as a matrix multiplication on the right. That is, any transformation from $\mathbb{R}^{m \times p} \mapsto \mathbb{R}^{m \times p}$ that can be expressed as $\boldsymbol{X} \mapsto \boldsymbol{X} \boldsymbol{M}$ for some $\boldsymbol{M} \in \mathbb{R}^{p \times p}$.

**Relation to Proposition 2** The space $\mathcal{H}$ corresponds to $\mathbb{S}^{m \times p}$ (the "sphere" of $m \times p$ matrices with unit Frobenius norm). The distance function $g$ is the angular distance, see eq. (4). The group $\mathcal{G}$ is defined as done directly above in our discussion of Proposition 1.

# C   Connections to Other Methods

This section describes the connections between generalized shape metrics and existing representational similarity measures in greater detail. For simplicity, we consider quantifying the similarity between two networks with $n$ neurons or hidden layer units. We use $\boldsymbol{X} \in \mathbb{R}^{m \times n}$ and $\boldsymbol{Y} \in \mathbb{R}^{m \times n}$ to denote matrices holding the hidden layer activations of two networks over $m$ common test inputs. In many cases, networks have distinct numbers of neurons or hidden units; however, this can be accommodated by applying PCA or zero-padding representations to achieve a common dimension.

For further simplicity, we will assume that $\boldsymbol{X}$ and $\boldsymbol{Y}$ are mean-centered such that $\boldsymbol{X}^\top \mathbf{1}_n = \boldsymbol{Y}^\top \mathbf{1}_n = \mathbf{0}_n$, where $\mathbf{0}_n$ and $\mathbf{1}_n$ respectively denote an $n$-dimensional vector of zeros and ones. Intuitively, this mean-centering removes the effect of translations in neural activation space when computing distances between neural representations. In the main text, we show this mean-centering step explicitly as a centering matrix $\boldsymbol{C} \in \mathbb{R}^{m \times m}$ that is included in the feature map, $\phi$. The mean-centering step is not strictly required, but is a typical preprocessing step in canonical correlations analysis [17] and Procrustes analysis [5].

## C.1   Permutation Invariance & Linear Assignment Problems

Consider the problem of finding the best permutation matrix which matches two sets of neural activations in terms of Euclidean distance. That is, we seek to find

$$\boldsymbol{\Pi}^* = \underset{\boldsymbol{\Pi} \in \mathcal{P}}{\operatorname{argmin}} \ \left\| \boldsymbol{X} - \boldsymbol{Y} \boldsymbol{\Pi} \right\|, \tag{23}$$

where $\mathcal{P}$ is the set of $n \times n$ permutation matrices. Note that this is equivalent to finding the permutation matrix that minimizes squared Euclidean distance, and that:

$$\left\| \boldsymbol{X} - \boldsymbol{Y} \boldsymbol{\Pi} \right\|^2 = \langle \boldsymbol{X}, \boldsymbol{X} \rangle + \langle \boldsymbol{Y}, \boldsymbol{Y} \rangle - 2 \langle \boldsymbol{X}, \boldsymbol{Y} \boldsymbol{\Pi} \rangle. \tag{24}$$

Since $\langle \boldsymbol{X}, \boldsymbol{X} \rangle$ and $\langle \boldsymbol{Y}, \boldsymbol{Y} \rangle$ are constant terms, the minimization in (23) is equivalent to:

$$\boldsymbol{\Pi}^* = \underset{\boldsymbol{\Pi} \in \mathcal{P}}{\operatorname{argmin}} \ -2 \langle \boldsymbol{X}, \boldsymbol{Y} \boldsymbol{\Pi} \rangle = \underset{\boldsymbol{\Pi} \in \mathcal{P}}{\operatorname{argmax}} \ \langle \boldsymbol{X}, \boldsymbol{Y} \boldsymbol{\Pi} \rangle = \underset{\boldsymbol{\Pi} \in \mathcal{P}}{\operatorname{argmin}} \ d_\theta(\boldsymbol{X}, \boldsymbol{Y} \boldsymbol{\Pi}). \tag{25}$$

The final equality holds since the angular distance is given by a monotonically decreasing function (i.e., $\arccos$) of the maximized inner product. Finally, using the definition of the Frobenius inner product, $\langle \boldsymbol{X}, \boldsymbol{Y} \boldsymbol{\Pi} \rangle = \operatorname{Tr}[\boldsymbol{X}^\top \boldsymbol{Y} \boldsymbol{\Pi}]$, and so,

$$\boldsymbol{\Pi}^* = \underset{\boldsymbol{\Pi} \in \mathcal{P}}{\operatorname{argmax}} \ \operatorname{Tr}[\boldsymbol{X}^\top \boldsymbol{Y} \boldsymbol{\Pi}]. \tag{26}$$

This final reformulation is the well-known *linear assignment problem* [1]. This can be solved efficiently in $O(n^3)$ time using standard algorithms [4], which are readily available in standard scientific computing environments. For example, the function `scipy.optimize.linear_sum_assignment` provides an implementation in Python [18].

## C.2   Orthogonal Procrustes Problems

Instead of optimizing over permutations, we may wish to optimize over orthogonal transformations. Given two matrices $\boldsymbol{X} \in \mathbb{R}^{m \times n}$ and $\boldsymbol{Y} \in \mathbb{R}^{m \times n}$, we seek to find

$$\boldsymbol{Q}^* = \operatorname*{argmin}_{\boldsymbol{Q} \in \mathcal{O}} \ \|\boldsymbol{X} - \boldsymbol{Y}\boldsymbol{Q}\|\,, \tag{27}$$

where $\mathcal{O}$ is the set of $n \times n$ orthogonal matrices. This is known as the orthogonal Procrustes problem [6]. Following the same steps as above in section C.1, we can see that $\boldsymbol{Q}^*$ also minimizes the angular distance between two matrices, and maximizes their inner product:

$$\boldsymbol{Q}^* = \operatorname*{argmax}_{\boldsymbol{Q} \in \mathcal{O}} \ \langle \boldsymbol{X}, \boldsymbol{Y}\boldsymbol{Q} \rangle = \operatorname*{argmin}_{\boldsymbol{Q} \in \mathcal{O}} \ d_\theta(\boldsymbol{X}, \boldsymbol{Y}\boldsymbol{Q})\,. \tag{28}$$

The following lemma states the well-known solution to this problem, which is due to Schönemann [14].

**Lemma 1** (Schönemann [14])**.** *Let $\boldsymbol{U}\boldsymbol{S}\boldsymbol{V}^\top$ denote the singular value decomposition of $\boldsymbol{X}^\top\boldsymbol{Y}$. Then $\boldsymbol{Q}^* = \boldsymbol{U}\boldsymbol{V}^\top$. Furthermore,*

$$\langle \boldsymbol{X}, \boldsymbol{Y}\boldsymbol{Q}^* \rangle = \|\boldsymbol{X}^\top\boldsymbol{Y}\|_* = \sum_i \sigma_i \tag{29}$$

*where $\|\cdot\|_*$ denotes the nuclear matrix norm and $\sigma_1 \geq \sigma_2, \geq \ldots \geq \sigma_n \geq 0$ are the singular values of $\boldsymbol{X}^\top\boldsymbol{Y}$.*

*Proof.* Let $\boldsymbol{Z} = \boldsymbol{V}^\top\boldsymbol{Q}\boldsymbol{U}$, and note that $\boldsymbol{Z}$ is orthogonal because orthogonal transformations are closed under composition. The cyclic property of the trace operator implies,

$$\max_{\boldsymbol{Q} \in \mathcal{Q}} \ \langle \boldsymbol{X}, \boldsymbol{Y}\boldsymbol{Q} \rangle = \max_{\boldsymbol{Q} \in \mathcal{Q}} \ \mathrm{Tr}[\boldsymbol{X}^\top\boldsymbol{Y}\boldsymbol{Q}] = \max_{\boldsymbol{Q} \in \mathcal{Q}} \ \mathrm{Tr}[\boldsymbol{S}\boldsymbol{V}^\top\boldsymbol{Q}\boldsymbol{U}] = \max_{\boldsymbol{Z} \in \mathcal{Q}} \ \mathrm{Tr}[\boldsymbol{S}\boldsymbol{Z}] = \max_{\boldsymbol{Z} \in \mathcal{Q}} \ \sum_{i=1}^n \sigma_i z_{ii} \tag{30}$$

where $\{z_{ii}\}_{i=1}^n$ are the diagonal elements of $\boldsymbol{Z}$. Since $\boldsymbol{Z}$ is orthogonal, we must have $z_{ii} \leq 1$ for all $i \in \{1, \ldots, n\}$. Since the singular values are nonnegative, the maximum is obtained when each $z_{ii} = 1$. That is, at optimality we have $\boldsymbol{Z} = \boldsymbol{V}^\top\boldsymbol{Q}^*\boldsymbol{U} = \boldsymbol{I}$, which implies $\boldsymbol{Q}^* = \boldsymbol{V}\boldsymbol{U}^\top$. Plugging $z_{ii} = 1$ into the final expression of eq. (30) shows that the optimal objective is given by the sum of the singular values (i.e. the nuclear norm of $\boldsymbol{X}^\top\boldsymbol{Y}$). $\qquad\square$

## C.3   Canonical Correlation Analysis (CCA)

CCA identifies matrices $\boldsymbol{W}_x \in \mathbb{R}^{n \times n}$ and $\boldsymbol{W}_y \in \mathbb{R}^{n \times n}$ which maximize the correlation between $\boldsymbol{X}\boldsymbol{W}_x$ and $\boldsymbol{Y}\boldsymbol{W}_y$. Formally, this corresponds to the optimization problem:

$$\begin{aligned} \operatorname*{maximize}_{\boldsymbol{W}_x, \boldsymbol{W}_y} \quad & \mathrm{Tr}[\boldsymbol{W}_x^\top\boldsymbol{X}^\top\boldsymbol{Y}\boldsymbol{W}_y] \\ \text{subject to} \quad & \boldsymbol{W}_x^\top\boldsymbol{X}^\top\boldsymbol{X}\boldsymbol{W}_x = \boldsymbol{W}_y^\top\boldsymbol{Y}^\top\boldsymbol{Y}\boldsymbol{W}_y = \boldsymbol{I}\,. \end{aligned} \tag{31}$$

The maximized objective function, $\langle \boldsymbol{XW}_x, \boldsymbol{YW}_y \rangle = \text{Tr}[\boldsymbol{W}_x^\top \boldsymbol{X}^\top \boldsymbol{YW}_y]$, generalizes the dot product between two vectors to the Frobenius inner product between $\boldsymbol{XW}_x$ and $\boldsymbol{YW}_y$. The constraints of the optimization problem constrain the magnitude of the solution—without these constraints, the objective function could be infinitely large, since multiplying $\boldsymbol{W}_x$ or $\boldsymbol{W}_y$ by a real number larger than one proportionally increases $\langle \boldsymbol{XW}_x, \boldsymbol{YW}_y \rangle$. Intuitively, the typical (Pearson) correlation is equal to the normalized inner product of two vectors, and CCA generalizes this to matrix-valued datasets.

CCA can be transformed into the Procrustes problem by a change of variables. Assuming that $\boldsymbol{X}^\top \boldsymbol{X}$ and $\boldsymbol{Y}^\top \boldsymbol{Y}$ are full rank, define $\boldsymbol{H}_x = (\boldsymbol{X}^\top \boldsymbol{X})^{1/2} \boldsymbol{W}_x$ and $\boldsymbol{H}_y = (\boldsymbol{Y}^\top \boldsymbol{Y})^{1/2} \boldsymbol{W}_y$. Then, (31) can be reformulated as:

$$\underset{\boldsymbol{H}_x, \boldsymbol{H}_y}{\text{maximize}} \quad \text{Tr}\left[ \boldsymbol{H}_x^\top (\boldsymbol{X}^\top \boldsymbol{X})^{-1/2} \boldsymbol{X}^\top \boldsymbol{Y} (\boldsymbol{Y}^\top \boldsymbol{Y})^{-1/2} \boldsymbol{H}_y \right]$$

$$\text{subject to} \quad \boldsymbol{H}_x^\top \boldsymbol{H}_x = \boldsymbol{H}_y^\top \boldsymbol{H}_y = \boldsymbol{I} \, . \tag{32}$$

By this change of variables, we simplified the constraints of the problem so that $\boldsymbol{H}_x$ and $\boldsymbol{H}_y$ are constrained to be orthogonal matrices. By applying the cyclic property of the trace operator, and defining $\boldsymbol{Q} = \boldsymbol{H}_y \boldsymbol{H}_x^\top$, $\boldsymbol{X}^\phi = \boldsymbol{X}(\boldsymbol{X}^\top \boldsymbol{X})^{-1/2}$, $\boldsymbol{Y}^\phi = \boldsymbol{Y}(\boldsymbol{Y}^\top \boldsymbol{Y})^{-1/2}$, we can simplify the problem further:

$$\underset{\boldsymbol{Q} \in \mathcal{O}}{\text{maximize}} \quad \text{Tr}[(\boldsymbol{X}^\phi)^\top \boldsymbol{Y}^\phi \boldsymbol{Q}] \, . \tag{33}$$

Thus, we see that CCA is equivalent to solving the Procrustes problem on $\boldsymbol{X}^\phi$ and $\boldsymbol{Y}^\phi$. Note that $(\boldsymbol{X}^\phi)^\top \boldsymbol{X}^\phi = (\boldsymbol{Y}^\phi)^\top \boldsymbol{Y}^\phi = \boldsymbol{I}$, and so this change of variables can be interpreted as a whitening operation [10].

From Lemma 1, we see that the optimal objective value to (33) is given by the sum of the singular values of $(\boldsymbol{X}^\phi)^\top \boldsymbol{Y}^\phi$. These singular values, which we denote here as $1 \geq \sigma_1 \geq ... \geq \sigma_n \geq 0$, are called *canonical correlation coefficients*. They are bounded above by one since the singular values of $\boldsymbol{X}^\phi$ and $\boldsymbol{Y}^\phi$ are all equal to one, due to the whitening step, and the operator norm[1] is sub-multiplicative:

$$\|(\boldsymbol{X}^\phi)^\top \boldsymbol{Y}^\phi\|_{\text{op}} \leq \|\boldsymbol{X}^\phi\|_{\text{op}} \|\boldsymbol{Y}^\phi\|_{\text{op}} = 1 \, . \tag{34}$$

Putting these pieces together, we see:

$$\underset{\boldsymbol{Q} \in \mathcal{O}}{\min} \ \arccos \frac{\langle \boldsymbol{X}^\phi, \boldsymbol{Y}^\phi \boldsymbol{Q} \rangle}{\|\boldsymbol{X}^\phi\| \|\boldsymbol{Y}^\phi\|} = \arccos \frac{\|(\boldsymbol{X}^\phi)^\top \boldsymbol{Y}^\phi\|_*}{\sqrt{n} \cdot \sqrt{n}} = \arccos \left( \frac{1}{n} \sum_{i=1}^{n} \sigma_i \right) \tag{35}$$

which coincides with equation 10 in the main text, since $\sigma_i = \rho_i / n$ for the case of CCA. Proposition 2 implies that this defines a metric since $\boldsymbol{X}^\phi / \|\boldsymbol{X}^\phi\|$ and $\boldsymbol{Y}^\phi / \|\boldsymbol{Y}^\phi\|$ are matrices with unit Frobenius norm, and because the set of orthogonal transformations is a group of isometries, as established in section A.5.

## C.4 Ridge CCA

Next, we consider metrics based on regularized CCA, which essentially interpolate between the orthogonally invariant metrics discussed in section C.2, and the linearly invariant metrics discussed in section C.3. This interpolation is accomplished by specifying a hyperparameter $0 \leq \alpha \leq 1$, where $\alpha = 0$ corresponds to unregularized CCA and $\alpha = 1$ corresponds to Procrustes alignment (i.e. fully regularized). We formulate this

---

[1]The operator norm of a matrix $\boldsymbol{M}$, denoted $\|\boldsymbol{M}\|_{\text{op}}$, is equal to the largest singular value of $\boldsymbol{M}$.

family of optimization problems as:

$$\underset{\boldsymbol{W}_x, \boldsymbol{W}_y}{\text{maximize}} \quad \text{Tr}[\boldsymbol{W}_x^\top \boldsymbol{X}^\top \boldsymbol{Y} \boldsymbol{W}_y]$$

$$\text{subject to} \quad \boldsymbol{W}_x^\top ((1-\alpha)\boldsymbol{X}^\top \boldsymbol{X} + \alpha \boldsymbol{I})\boldsymbol{W}_x = \boldsymbol{W}_y^\top ((1-\alpha)\boldsymbol{Y}^\top \boldsymbol{Y} + \alpha \boldsymbol{I})\boldsymbol{W}_y = \boldsymbol{I}\,. \tag{36}$$

Notice that when $\alpha = 1$, the constraints reduce to $\boldsymbol{W}_x$ and $\boldsymbol{W}_y$ being orthogonal, and thus the objective function can be viewed as maximizing $\langle \boldsymbol{X}, \boldsymbol{Y}\boldsymbol{Q}\rangle$ over orthogonal matrices $\boldsymbol{Q} = \boldsymbol{W}_y \boldsymbol{W}_x^\top$. Thus, we recover Procrustes alignment in the limit of $\alpha = 1$. Clearly, when $\alpha = 0$, eq. (36) reduces to the usual formulation of CCA (see eq. (31)).

We can solve eq. (36) by following essentially the same procedure outlined in section C.3, in which we reduce the problem to Procrustes alignment by a change of variables. In this case, the change of variables corresponds to a partial whitening transformation:

$$\boldsymbol{H}_x = ((1-\alpha)(\boldsymbol{X}^\top \boldsymbol{X}) + \alpha \boldsymbol{I}))^{1/2} \quad \text{and} \quad \boldsymbol{H}_y = ((1-\alpha)(\boldsymbol{Y}^\top \boldsymbol{Y}) + \alpha \boldsymbol{I}))^{1/2}\,. \tag{37}$$

Then, reformulate the optimization problem as:

$$\underset{\boldsymbol{H}_x, \boldsymbol{H}_y}{\text{maximize}} \quad \text{Tr}[\boldsymbol{H}_x^\top ((1-\alpha)(\boldsymbol{X}^\top \boldsymbol{X}) + \alpha \boldsymbol{I}))^{-1/2} \boldsymbol{X}^\top \boldsymbol{Y} ((1-\alpha)(\boldsymbol{Y}^\top \boldsymbol{Y}) + \alpha \boldsymbol{I}))^{-1/2} \boldsymbol{H}_y]$$

$$\text{subject to} \quad \boldsymbol{H}_x^\top \boldsymbol{H}_x = \boldsymbol{H}_y^\top \boldsymbol{H}_y = \boldsymbol{I}\,. \tag{38}$$

Let $\boldsymbol{Q} = \boldsymbol{H}_y \boldsymbol{H}_x$, and let

$$\boldsymbol{X}^\phi = \boldsymbol{X}((1-\alpha)(\boldsymbol{X}^\top \boldsymbol{X}) + \alpha \boldsymbol{I}))^{-1/2} \quad \text{and} \quad \boldsymbol{Y}^\phi = \boldsymbol{Y}((1-\alpha)(\boldsymbol{Y}^\top \boldsymbol{Y}) + \alpha \boldsymbol{I}))^{-1/2}\,. \tag{39}$$

Then, by Proposition 2 and lemma 1, we have the following metric:

$$\underset{\boldsymbol{Q}\in\mathcal{O}}{\min} \ \arccos \frac{\langle \boldsymbol{X}^\phi, \boldsymbol{Y}^\phi \boldsymbol{Q}\rangle}{\|\boldsymbol{X}^\phi\|\|\boldsymbol{Y}^\phi\|} = \arccos \left\| \left(\frac{\boldsymbol{X}^\phi}{\|\boldsymbol{X}^\phi\|}\right)^\top \left(\frac{\boldsymbol{Y}^\phi}{\|\boldsymbol{Y}^\phi\|}\right) \right\|_* = \arccos \left(\sum_{i=1}^n \rho_i\right)\,. \tag{40}$$

## C.5 Nonlinear Alignments and Kernel CCA

We can also consider metrics based on *kernel CCA* [7], which generalizes CCA to account for nonlinear alignments. As its name suggests, this approach belongs to a more general class of *kernel methods* that operate implicitly in high-dimensional (even infinite-dimensional) feature spaces through inner product evaluations. For a broader review of kernel methods in machine learning, see [9].

First, we recall the inner product between two matrices in a finite dimensional feature space $\mathbb{R}^{m\times p}$:

$$\langle \boldsymbol{X}^\phi, \boldsymbol{Y}^\phi\rangle = \text{Tr}[(\boldsymbol{X}^\phi)^\top \boldsymbol{Y}^\phi] = \sum_{i=1}^m (\boldsymbol{x}_i^\phi)^\top (\boldsymbol{y}_i^\phi)\,. \tag{41}$$

Here we have introduced notation $\boldsymbol{x}_i^\phi$ and $\boldsymbol{y}_i^\phi$ to denote the $p$-dimensional vectors holding features to the $i^{\text{th}}$ network input. In kernel CCA, we consider more general feature mappings $\boldsymbol{x}_i \mapsto x_i^\phi$ and $\boldsymbol{y}_i \mapsto y_i^\phi$, where each $x_i^\phi \in \mathcal{H}$ and $y_i^\phi \in \mathcal{H}$ are vectors in some Reproducing Kernel Hilbert Space (RKHS). That is, instead of having two matrices $\boldsymbol{X}^\phi$ and $\boldsymbol{Y}^\phi$ to represent the network representations in the feature space, we instead consider the collections of vectors: $X^\phi = \{x_1^\phi, \ldots, x_m^\phi\}$ and $Y^\phi = \{y_1^\phi, \ldots, y_m^\phi\}$.

Given a choice of a positive-definite kernel function $k$, we begin by computing two $m \times m$ un-centered kernel matrices:

$$[\widetilde{\boldsymbol{K}}_x]_{ij} = k(\boldsymbol{x}_i, \boldsymbol{x}_j) = \langle x_i^\phi, x_j^\phi \rangle \quad \text{and} \quad [\widetilde{\boldsymbol{K}}_y]_{ij} = k(\boldsymbol{y}_i, \boldsymbol{y}_j) = \langle y_i^\phi, y_j^\phi \rangle \tag{42}$$

for $i, j \in \{1, \dots, m\}$. Then, we define the centered kernel matrices: $\boldsymbol{K}_x = \boldsymbol{C}\widetilde{\boldsymbol{K}}_x\boldsymbol{C}$ and $\boldsymbol{K}_y = \boldsymbol{C}\widetilde{\boldsymbol{K}}_y\boldsymbol{C}$, where $\boldsymbol{C} = \boldsymbol{I} - \frac{1}{m}\boldsymbol{1}\boldsymbol{1}^\top$ is the centering matrix.

The classic form of CCA (31) can then be reformulated terms of purely kernel operations [7, 17]:

$$\begin{aligned} &\underset{\boldsymbol{W}_x, \boldsymbol{W}_y}{\text{maximize}} \quad \text{Tr}\left[\boldsymbol{W}_x^\top \boldsymbol{K}_x \boldsymbol{K}_y \boldsymbol{W}_y\right] \\ &\text{subject to} \quad \boldsymbol{W}_x^\top \boldsymbol{K}_x^2 \boldsymbol{W}_x = \boldsymbol{W}_y^\top \boldsymbol{K}_y^2 \boldsymbol{W}_y = \boldsymbol{I}. \end{aligned} \tag{43}$$

One can show that this optimization problem is equivalent (up to a change of variables) from the classic CCA problem when a linear kernel function, $k(\boldsymbol{x}_i, \boldsymbol{x}_j) = \boldsymbol{x}_i^\top \boldsymbol{x}_j$, is used. Furthermore, one can generalize the regularization scheme for CCA (see section C.4),

$$\begin{aligned} &\underset{\boldsymbol{W}_x, \boldsymbol{W}_y}{\text{maximize}} \quad \text{Tr}\left[\boldsymbol{W}_x^\top \boldsymbol{K}_x \boldsymbol{K}_y \boldsymbol{W}_y\right] \\ &\text{subject to} \quad \boldsymbol{W}_x^\top ((1-\alpha)\boldsymbol{K}_x^2 + \alpha\boldsymbol{K}_x)\boldsymbol{W}_x = \boldsymbol{W}_y^\top ((1-\alpha)\boldsymbol{K}_y^2 + \alpha\boldsymbol{K}_y)\boldsymbol{W}_y = \boldsymbol{I}. \end{aligned} \tag{44}$$

## C.6 Geodesic Distances on Kendall's Shape Space

We now consider a modification of the Procrustes alignment problem, where we optimize over the special orthogonal group (i.e. the set of orthogonal matrices with $\det(\boldsymbol{Q}) = +1$)

$$\boldsymbol{R}^* = \underset{\boldsymbol{R} \in \mathcal{SO}}{\operatorname{argmin}} \ \|\boldsymbol{X} - \boldsymbol{Y}\boldsymbol{R}\| = \underset{\boldsymbol{R} \in \mathcal{SO}}{\operatorname{argmax}} \ \text{Tr}[\boldsymbol{X}^\top \boldsymbol{Y} \boldsymbol{R}]. \tag{45}$$

We can obtain the solution by a minor modification of Lemma 1. We let $\boldsymbol{X}^\top \boldsymbol{Y} = \tilde{\boldsymbol{U}}\tilde{\boldsymbol{S}}\tilde{\boldsymbol{V}}^\top$ denote the "optimally signed" singular value decomposition of $\boldsymbol{X}^\top \boldsymbol{Y}$ in which $\tilde{\boldsymbol{U}} \in \mathcal{SO}$, $\tilde{\boldsymbol{V}} \in \mathcal{SO}$, and $\tilde{\boldsymbol{S}}$ is a diagonal matrix of signed singular values: $\tilde{\sigma}_1 \geq \dots \geq \tilde{\sigma}_{n-1} \geq |\tilde{\sigma}_n| \geq 0$. Thus, all optimally signed singular values are positive except if $\det(\boldsymbol{X}^\top \boldsymbol{Y}) < 0$, in which case the final singular value is negated, $\tilde{\sigma}_n = -\sigma_n$, so that $\det(\tilde{\boldsymbol{U}}) = \det(\tilde{\boldsymbol{V}}) = +1$. Then the optimal rotation is given by $\boldsymbol{R}^* = \tilde{\boldsymbol{V}}\tilde{\boldsymbol{U}}^\top$. See Le [13] for a proof.

We refer the reader to Chapters 4 and 5 of Dryden and Mardia [5] for further details. When $\mathcal{G} = \mathcal{SO}$, our Proposition 1 corresponds to Riemannian distance in size-and-shape space (sec. 5.3, [5]). Likewise, $\mathcal{G} = \mathcal{SO}$, our Proposition 2 corresponds to Riemannian distance Kendall's shape space (sec. 4.1.4, [5]).

## C.7 Centered Kernel Alignment (CKA) and Representational Similarity Analysis (RSA)

Linear CKA [11] and RSA [12] are two closely related methods that, in essence, evaluate the similarity between $\boldsymbol{X}\boldsymbol{X}^\top$ and $\boldsymbol{Y}\boldsymbol{Y}^\top$ to capture the similarity of neural representations. When the data are mean-centered as a preprocessing step, these are $m \times m$ covariance matrices capturing the correlations in neural activations over the $m$ test images. Several variants of RSA exist. For example, one can compute the pairwise Euclidean distances between all $m$ hidden layer activation patterns, resulting in representational distance matrices (RDMs) instead of the covariance matrices mentioned above. Likewise, nonlinear extensions of CKA use nonlinear kernel functions to compute centered kernel matrices $\boldsymbol{K}_x$ and $\boldsymbol{K}_y$, as defined above in

section C.5. When a linear kernel function is used (i.e. in linear CKA), the centered kernel matrices reduce to the usual covariance matrices $\boldsymbol{K}_x = \boldsymbol{X}\boldsymbol{X}^\top$ and $\boldsymbol{K}_y = \boldsymbol{Y}\boldsymbol{Y}^\top$.

In essence, these methods proceed by computing the similarity between $\boldsymbol{K}_x$ and $\boldsymbol{K}_y$. Kriegeskorte et al. [12] proposed taking the Spearman correlation between the upper-triangular entries of these matrices. This measure of similarity does not produce a metric, as we verified empirically in the main text. Kornblith et al. [11] proposed to use the following quantity (assuming centered kernels):

$$\mathrm{CKA}(\boldsymbol{K}_x, \boldsymbol{K}_y) = \frac{\mathrm{Tr}[\boldsymbol{K}_x \boldsymbol{K}_y]}{\sqrt{\mathrm{Tr}[\boldsymbol{K}_x^2] \cdot \mathrm{Tr}[\boldsymbol{K}_y^2]}} \tag{46}$$

which is known as centered kernel alignment (originally defined in [3, 2]).

While CKA as originally formulated does not produce a metric, we can modify it to satisfy the requirements of a metric space. First, note that:

$$\mathrm{CKA}(\boldsymbol{K}_x, \boldsymbol{K}_y) = \cos\left[d_\theta(\boldsymbol{K}_x, \boldsymbol{K}_y)\right] \tag{47}$$

where $d_\theta$ is the angular distance (see eq. (4)) over $\mathbb{R}^{m \times m}$ matrices. Thus, one can apply $\arccos(\cdot)$ to CKA achieve a proper metric. For example, a metric based on linear CKA can be calculated as follows:

$$d_\theta(\boldsymbol{X}\boldsymbol{X}^\top, \boldsymbol{Y}\boldsymbol{Y}^\top) = \arccos\left[\frac{\|\boldsymbol{X}^\top \boldsymbol{Y}\|^2}{\|\boldsymbol{X}\boldsymbol{X}^\top\|\|\boldsymbol{Y}\boldsymbol{Y}^\top\|}\right] \tag{48}$$

where, as before, all norms denote the Frobenius matrix norm. Note that this calculation bears some similarity to the fully regularized CCA distance:

$$\theta_1(\boldsymbol{X}, \boldsymbol{Y}) = \min_{\boldsymbol{Q} \in \mathcal{O}} \arccos\left[\frac{\langle \boldsymbol{X}, \boldsymbol{Y}\boldsymbol{Q} \rangle}{\|\boldsymbol{X}\| \cdot \|\boldsymbol{Y}\|}\right] = \arccos\left[\frac{\|\boldsymbol{X}^\top \boldsymbol{Y}\|_*}{\|\boldsymbol{X}\|\|\boldsymbol{Y}\|}\right] \tag{49}$$

The two differences between these metrics are that (a) CKA uses the squared Frobenius norm instead of the nuclear norm to measure the scale of $\boldsymbol{X}^\top \boldsymbol{Y}$ in the numerator, and (b) CKA normalizes by the norms of the covariances, $\boldsymbol{X}\boldsymbol{X}^\top$ and $\boldsymbol{Y}\boldsymbol{Y}^\top$, rather than the norms of the matrices themselves.

While this manuscript was undergoing review, Shahbazi et al. [15] published a different modification of CKA and RSA to satisfy the properties of a metric space. They advocate using the Riemannian metric over positive-definite matrices:

$$d(\boldsymbol{K}_x, \boldsymbol{K}_y) = \sqrt{\sum_{i=1}^m \log^2(\lambda_i)}, \tag{50}$$

where $\lambda_1, \ldots, \lambda_m$ are the eigenvalues of $\boldsymbol{K}_x^{-1} \boldsymbol{K}_y$. This calculation is appealing because it exploits the fact that $\boldsymbol{K}_x$ and $\boldsymbol{K}_y$ are positive-definite matrices by construction. The extension of CKA discussed above utilizes the generic angular distance between $m \times m$ matrices, which are not necessarily positive-definite.

# D  Probabilistic interpretations of generalized shape metrics

To extend generalized shape metrics to stochastic neural representations, we must introduce some additional notation and formalize network representations as random variables (rather than $m \times n$ matrices). We can

model neural representations as independent random variables when conditioned on the input. That is, let $X$ and $Y$ denote random variables on $\mathbb{R}^n$, which correspond to $n$-dimensional neural responses to a stochastic input.[2] Further, let $Z$ be some random variable corresponding to process of sampling an input to the network (e.g. choosing one of $m$ input images at random). Then, the joint distribution over representations and inputs decomposes as $P(X, Y, Z) = P(X \mid Z)P(Y \mid Z)P(Z)$ for any pair of networks $X$ and $Y$.

The goal of this section is to define functions $d(X, Y)$ that are metrics over the set of random variables with outcomes on $\mathbb{R}^n$, and which are natural extensions of Proposition 1 and 2 in the main text. The key step towards achieving this goal is to establish a Hilbert space for random vectors. We provide a short and informal demonstration of this below, but refer the reader to Chapter 2 of Tsiatis [16] for a more complete treatment.

First, we establish that the set of random vectors is a vector space. The zero vector corresponds to a random vector that is equal to the zero vector on $\mathbb{R}^n$ almost surely. Vector addition $X + Y$ creates a new random vector from two inputs $X$ and $Y$. Intuitively, we can draw samples from $X + Y$ by first sampling $X$ and $Y$ and then adding their outcomes. Scalar multiplication $\alpha X$ creates a new random vector given the input $X$ and a scalar $\alpha \in \mathbb{R}$. Intuitively, we can sample $\alpha X$ by first drawing a sample from $X$ and multiplying this outcome by $\alpha$. We can then define the inner product between two random vectors in the following lemma.

**Lemma.** *Let $X$ and $Y$ be random vectors associated with some joint probability density function $p(\boldsymbol{x}, \boldsymbol{y})$ for all $\boldsymbol{x} \in \mathbb{R}^n$ and $\boldsymbol{y} \in \mathbb{R}^n$. Then,*

$$\langle X, Y \rangle = \mathbb{E}[\boldsymbol{x}^\top \boldsymbol{y}], \tag{51}$$

*is an inner product over the set of random vectors, where the expectation is taken over joint samples of $X$ and $Y$.*

*Proof.* Using the linearity of expectation and the inner product on $\mathbb{R}^n$, it is easy to prove that the inner product is symmetric,

$$\langle X, Y \rangle = \mathbb{E}[\boldsymbol{x}^\top \boldsymbol{y}] = \mathbb{E}[\boldsymbol{y}^\top \boldsymbol{x}] = \langle Y, X \rangle, \tag{52}$$

and linear,

$$\langle M + \alpha X, Y \rangle = \mathbb{E}[(\boldsymbol{z} + \alpha \boldsymbol{x})^\top \boldsymbol{y}] = \mathbb{E}[\boldsymbol{z}^\top \boldsymbol{y}] + \alpha \mathbb{E}[\boldsymbol{x}^\top \boldsymbol{y}] = \langle M, Y \rangle + \alpha \langle X, Y \rangle \tag{53}$$

for any random vector $M$ and $\alpha \in \mathbb{R}$. All that remains is to prove is that $\langle \cdot, \cdot \rangle$ is positive definite, we first note that the mapping $\boldsymbol{x} \mapsto \boldsymbol{x}^\top \boldsymbol{x}$ is a convex function of $\boldsymbol{x}$. Then, we apply Jensen's inequality and the positive definiteness of the inner product on $\mathbb{R}^n$ to show:

$$\langle X, X \rangle = \mathbb{E}[\boldsymbol{x}^\top \boldsymbol{x}] \geq (\mathbb{E}\boldsymbol{x})^\top (\mathbb{E}\boldsymbol{x}) \geq 0. \tag{54}$$

Further $\mathbb{E}[\boldsymbol{x}^\top \boldsymbol{x}] = 0$ only when $\boldsymbol{x} = \boldsymbol{0}$, almost surely. Thus, $\langle X, X \rangle = 0$ if and only if $X = 0$. $\square$

To begin, we consider a special case where the neural responses are deterministic, but the inputs are randomly chosen. That is, to draw a sample of $(X, Y)$, we first sample an input $\boldsymbol{z} \sim P(Z)$ and then calculate $\boldsymbol{x} = f_x(\boldsymbol{z})$ and $\boldsymbol{y} = f_y(\boldsymbol{z})$, where $f_x$ and $f_y$ are functions mapping the input space to $\mathbb{R}^n$.

In the simplest case, $P(Z)$ is a uniform distribution over a discrete set of $m$ network inputs. In this case, we can compute the required inner products exactly. Let $\boldsymbol{z}_i$ denote the $i^{\text{th}}$ input to the networks, and let

---

[2]As in the main text, we can define feature maps $X \mapsto X^\phi$ and $Y \mapsto Y^\phi$ which establish a common dimensionality between networks of dissimilar sizes.

$\boldsymbol{X} \in \mathbb{R}^{m \times n}$ and $\boldsymbol{Y} \in \mathbb{R}^{m \times n}$ denote matrices that stack the neural responses, $f_x(\boldsymbol{z}_i)$ and $f_y(\boldsymbol{z}_i)$ row-wise. Then we have

$$\langle X, Y \rangle = \mathbb{E}[\boldsymbol{x}^\top \boldsymbol{y}] = \frac{1}{m} \sum_{i=1}^{m} f_x(\boldsymbol{z}_i)^\top f_y(\boldsymbol{z}_i) = \frac{1}{m} \langle \boldsymbol{X}, \boldsymbol{Y} \rangle, \tag{55}$$

where the final inner product $\langle \boldsymbol{X}, \boldsymbol{Y} \rangle = \text{Tr}[\boldsymbol{X}^\top \boldsymbol{Y}]$ is the typical Frobenius inner product between matrices that we have used throughout. Because these inner products coincide up to a uniform scaling factor, we can reinterpret the metrics defined in the main text (Propositions 1 & 2) as providing a notion of distance between deterministic neural responses that are drawn uniformly from a set of $m$ inputs.

In many cases, the number of possible inputs to a network is effectively infinite, so we can consider $P(Z)$ to be a continuous distribution. In this scenario, the inner product becomes:

$$\langle X, Y \rangle = \int p(\boldsymbol{z}) f_x(\boldsymbol{z})^\top f_y(\boldsymbol{z}) \, \mathrm{d}\boldsymbol{z} \tag{56}$$

which is generally intractable to compute. For example, we typically do not know how to evaluate the density $p(\boldsymbol{z})$. This is the case, for example, when $P(Z)$ corresponds to the distribution over all "natural images." If we are given independent samples $\boldsymbol{z}_i \sim P(Z)$, for $i = 1, \ldots, m$, then the integral can be approximated as

$$\int p(\boldsymbol{z}) f_x(\boldsymbol{z})^\top f_y(\boldsymbol{z}) \, \mathrm{d}\boldsymbol{z} \approx \frac{1}{m} \sum_{i=1}^{m} f_x(\boldsymbol{z}_i)^\top f_y(\boldsymbol{z}_i) = \frac{1}{m} \langle \boldsymbol{X}, \boldsymbol{Y} \rangle, \tag{57}$$

which coincides with (55). Thus, we can also interpret generalized shape metrics (Propositions 1 & 2) as being approximations to metrics that capture representational dissimilarity over a continuous distribution of input patterns. This final interpretation is appealing from both scientific and engineering perspectives. In neuroscience, we expect animals to encounter sensory input patterns probabilistically from an effectively infinite range of possibilities. Likewise, in machine learning, we are interested in how deep artificial networks generalize to "real-world" applications. In short, the space of possible future inputs is generally more numerous than the space of inputs used for training and validation. Nonetheless, if the statistics of the test set match the "real world," then (57) tells us that we can approximate the "true" distance between network representations appropriately.

The results shown in Figures 3B and 3C in the main text can now be properly interpreted as varying the choice of $m$ (sample size) in the approximation of the integral appearing in equation (57).

The framework above can also be readily extended to define metrics between stochastic neural representations, which are ubiquitous in both biology (due to "noise") and machine learning (e.g. dropout layers). We view this as an intriguing direction for future research that is enabled by our theoretical framing of neural representations.

# E   Experimental Methods

Code accompanying this paper can be found at — https://github.com/ahwillia/netrep

## E.1   Experiments on sample size (Fig. 3)

We ran all experiments on a pair of convolutional neural networks trained on CIFAR-10. The architecture is shown in Table 1. In Figure 3A, we sampled activations from the three layers following the stride-2

convolutions. We did a brute-force search over circular shifts along the width and height dimensions. When comparing two layers with unequal dimensions, we upsampled the layer with smaller width and height by linear interpolation. The remaining panels in Figure 3 were computed using activations from the final layer before average pooling.

| |
| --- |
| $3 \times 3$ conv. 64-BN-ReLU |
| $3 \times 3$ conv. 64-BN-ReLU |
| $3 \times 3$ conv. 64-BN-ReLU |
| $3 \times 3$ conv. 64 stride 2-BN-ReLU |
| $3 \times 3$ conv. 128-BN-ReLU |
| $3 \times 3$ conv. 128-BN-ReLU |
| $3 \times 3$ conv. 128-BN-ReLU |
| $3 \times 3$ conv. 128 stride 2-BN-ReLU |
| $3 \times 3$ conv. 256-BN-ReLU |
| $3 \times 3$ conv. 256-BN-ReLU |
| $3 \times 3$ conv. 256-BN-ReLU |
| $3 \times 3$ conv. 256 stride 2-BN-ReLU |
| Global average pooling |
| Logits |

*Table 1:* The architecture used for experiments in Fig. 3. All convolutions use zero padding to maintain the size of the feature map.

## E.2 Allen Brain Observatory

Data were accessed through the Allen Software Development Kit (AllenSDK — https://allensdk.readthedocs.io/en/latest/). All isolated single units that met the default quality control standards were loaded and pooled across sessions. The anatomical location of each unit in Common Coordinate Framework (CCF; [19]) was extracted and categorized into anatomical regions according to the reference atlas, using the finest scale anatomical parcellation. Spike counts were calculated over 0.033355 ms timebins (duration of a single movie frame), over 1600 frames. Spikes were then smoothed with a Gaussian filter with a standard deviation of 20 bins (frames), and averaged over 10 trials (repeats of the movie). Then, we projected the data onto the top 100 principal components, resulting in a matrix $X_k \in \mathbb{R}^{1600 \times 100}$ for each brain region $k = \{1, \ldots, K\}$. Regions with fewer than 100 neurons across all sessions were excluded. The following set of 48 regions, listed by their standard abbreviations, contained more than 100 neurons and were then studied for further analysis: APN, AUDd5, AUDpo5, AUDpo6a, CA1, CA3, DG-mo, DG-sg, Eth, LGd-co, LGd-ip, LGd-sh, LGv, LP, MB, MGd, MGv, PO, POL, ProS, SGN, SSp-bfd2/3, SSp-bfd4, SSp-bfd5, SUB, TEa5, TH, VISa2/3, VISa4, VISa5, VISa6a, VISal2/3, VISal4, VISal5, VISam2/3, VISam4, VISam5, VISam6a, VISp2/3, VISp4, VISp5, VISp6a, VPM, alv, ccs, dhc, fp, or.

Dendrograms were computed and visualized using tools available in the scipy library [18]. We used Ward's linkage criterion to compute the hierarchical clusterings.

We performed kernel ridge regression to predict anatomical hierarchy scores (defined in [8]) 29 regions: AUDd5, AUDpo5, AUDpo6a, LGd-co, LGd-ip, LGd-sh, LP, MGd, MGv, POL, SSp-bfd2/3, SSp-bfd4, SSp-bfd5, TEa5, VISa2/3, VISa4, VISa5, VISa6a, VISal2/3, VISal4, VISal5, VISam2/3, VISam4, VISam5, VISam6a, VISp2/3, VISp4, VISp5, VISp6a. Two regions, PO and VPM, were excluded from the analysis as they were outliers with exceptionally high and low hierarchy scores. The other regions were excluded because they either had undefined hierarchy scores or had fewer than 100 neurons. We used the scikit-learn implementation

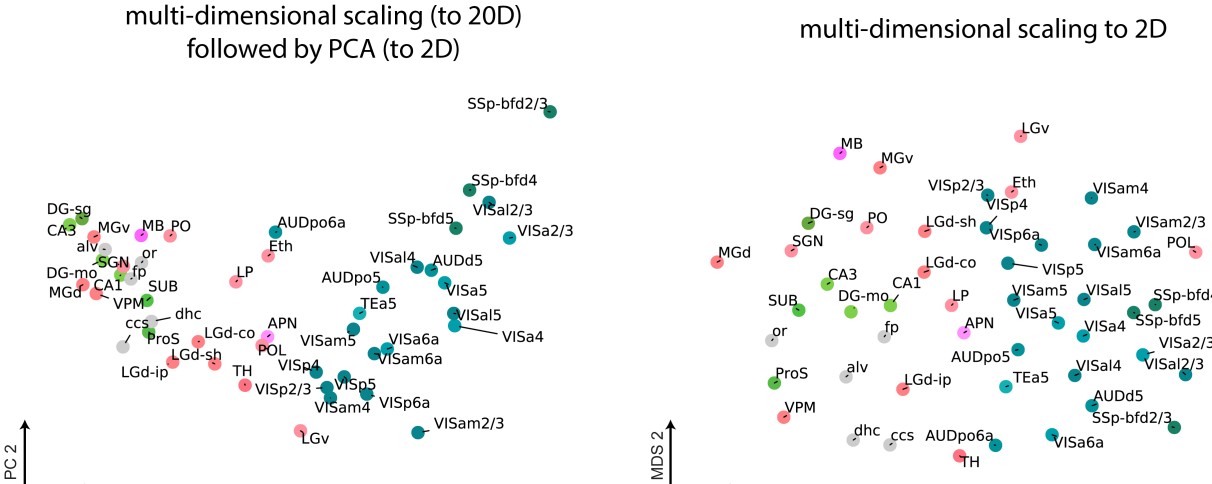

*Figure E.2.1:* Performing MDS directly to $L = 2$ dimensions (right) produces a distinct low-dimensional visualization of the ABO dataset from multi-dimensional scaling to $L = 20$ dimensions, following by PCA projection down to 2D (left). As shown in the main text (Fig. 4C), the MDS embedding to $L = 20$ dimensions produces a dramatically better approximation of the true metric space than the embedding to $L = 2$ dimensions. Thus, we advocate using the former over the latter for downstream modeling tasks.

of kernel ridge regression, `KernelRidge(alpha=0.01, gamma=1.0, kernel="rbf")`, and fit the model 100 separate times on different approximate Euclidean embeddings found by multi-dimensional scaling (MDS). The error bars in Fig. 5B show range of estimates from different MDS embeddings. An embedding dimension of $L = 20$ was used in all cases.

If our goal is only to visualize the data in 2D we may apply MDS with an embedding dimension of $L = 2$. How does this embedding differ from a larger embedding of $L = 20$? Figure E.2.1 demonstrates that qualitatively distinct structures emerge from these two procedures.

## E.3 NAS-Bench-101

We obtained checkpoints for 2000 randomly-selected NAS-Bench-101 architectures trained for 108 epochs following the protocol described in [20] and computed the similarity between activations of every possible pair of these architectures on the CIFAR-10 test set, using an Apache Beam pipeline operating on offsite hardware. In total, the computational cost of these experiments was 260 core-years, including pilot experiments and several experiments not included in the paper.

For ridge regression analyses in Fig. 5D, we train on 80% of the data, use 10% of the data as a validation set to select the optimal ridge hyperparameter and the kernel bandwidth, and compute $R^2$ on remaining 10% of the data.

In Figure E.3.1, we show the skeleton of the NAS-Bench-101 architecture along with the layers from which we extract representations.

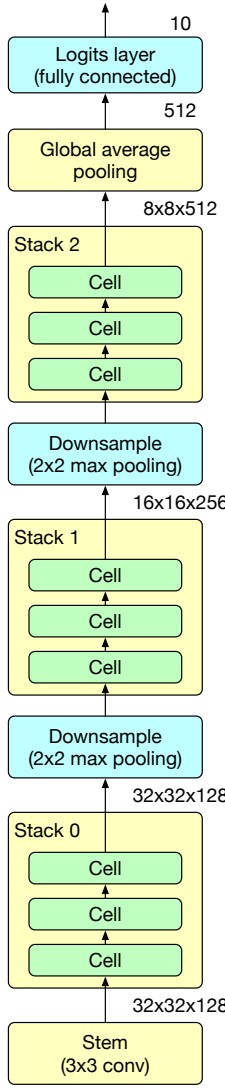

*Figure E.3.1:* Diagram of the skeleton of the NAS-Bench-101 architecture. The architecture of each cell (shown in green) is selected from a fixed space, described further by Ying et al. [20], and all cells within a single architecture are identical except for the number of channels, which differs by stack. In Fig. 5, we show the results we obtain by analyzing the representations of the outputs of the layers shown in yellow.