# OpenReview forum: "Generalized Shape Metrics on Neural Representations"
_NeurIPS.cc/2021/Conference — NeurIPS 2021 Poster_

### Official Review · Reviewer_Fmhp · 2021-07-16

**Rating:** 6
**Confidence:** 4

**Summary:**

This paper develops metrics for comparing different (yet presumably related) representations. The emphasis is on representations generated by artificial or biological neural networks. The paper relates this development to previously proposed approaches. Experimental results on recordings of neuronal activity in the mouse brain and representations generated by different layers of different deep neural networks are presented.

**Limitations And Societal Impact:**

No, but I don't think this mathematical development carries foreseeable potential negative societal impact.

**Main Review:**

- It is not clear why representations of different networks should be linearly related. It seems identifying topological similarity would be a better fit for this problem, although it may be computationally impractical. I find the reasoning behind the equivalence classes being defined through linear transformations missing. At least, a topology-centered introduction which discusses its practicality before settling on linear transformations would be a better exposition.
- It takes a while to realize that this work concerns the alignment of representations and these representations do not have to come from networks. Network representations is the application here, although the development does not have much to do with having an underlying biological/artificial neural network.
- The paper devotes ~5 pages to reviewing the basics and setting up the problem. As a result, the experimental section is somewhat rushed and whether the approach offers practical improvements is not clear. Although, of course, having a principled approach is inherently advantageous.
-  On a related note, I couldn’t tell if the authors claim novelty about the propositions. They are rather elementary, including the generalized version in the supplement, which covers both cases. It wouldn’t surprise me if they appear in a textbook etc although I can’t cite one here. Please clarify this point.
- The experiments would be much more meaningful in the presence of baselines. For instance, if the ABO experiment were repeated, except with non-metric embeddings, but following the same multidimensional scaling and PCA parameters, how different would the embedding of the brain regions be?
- Why does it make a big difference to embed in L-dim space first and then perform dim. reduction rather than embedding immediately?

Minor:
- (line 120) Please define the equivalence relation.
- (line 31) As written, the definition corresponds to a semi-metric. Please include the equivalence condition here as well.

Clarity: The exposition is generally clear and it is not hard to follow the development.

------------------------------ UPDATE ------------------------------

I decided to increase my score from 5 to 6. Please refer to the discussion with the authors for details on the topological aspects of the problem.

**Time Spent Reviewing:**

7

---

> ### Author Response · Authors · 2021-08-10
> **Response**
>
> We thank the reviewer for their feedback and careful reading of the manuscript. We believe that we can address all of their concerns with minor edits to the manuscript which will make our contributions much more clear and improve the final paper.
>
> > *"It is not clear why representations of different networks should be linearly related."*
>
> Generalized shape metrics actually do allow for nonlinear alignments and notions of equivalence, since the feature map, $\phi$, may be nonlinear. We outline a connection to kernel CCA in the Supplement and mention it briefly in the main text. We will highlight this important point in the final paper.
>
> Nonetheless, in practice, people have generally found linear or orthogonal transformations to be the most useful and interpretable, e.g. refs 4-7, 10, 11. This may be because the nonlinear alignments add non-trivial additional structure to the representations. The discussion of mutual information as a similarity criterion in Kornblith et al. (2019, "Similarity of Neural Network Representations Revisited") makes a compelling argument for this somewhat counterintuitive phenomenon. We will discuss this point in greater depth and add a more direct reference to Kornblith et al. (2019) in our edits for the camera-ready manuscript.
>
> Given the limited space, we chose to focus on applications of linear shape metrics in our manuscript. However, we agree that a deeper investigation of nonlinear shape metrics are an important direction for future work.
>
> > *"It seems identifying topological similarity would be a better fit for this problem, although it may be computationally impractical..."*
>
> As discussed above, linear similarity measures are ubiquitous in current deep learning and neuroscience literature. Thus, while we find the idea of motivating the paper from a topological perspective interesting, we believe it merits a separate investigation. Our current work aims to build on existing approaches by grounding them in formal metric spaces and then exploiting this metric space structure to enable new analyses (e.g. supervised learning via approximate Euclidean embeddings of hidden representations). These approaches are built on linear similarity are extendable to nonlinear similarity through kernel methods.
>
> > *"Network representations is the application here, although the development does not have much to do with having an underlying biological/artificial neural network."*
>
> We will edit the introduction to make it clear that our theoretical results have broader applicability.
>
> We note that section 2.4 does develop a framework that is specialized to convolutional layers, and that this framework has demonstrated benefits over existing heuristic ("flattened") representations in terms of the amount of data one needs to estimate the representational distances (see Fig. 3B). So some of our results are adapted specifically to the artifical network case.
>
> > "As a result, the experimental section is somewhat rushed and whether the approach offers practical improvements is not clear. Although, of course, having a principled approach is inherently advantageous."
>
> We clarify the practical improvements we introduce in a detailed comment below.
>
> We agree that a principled approach is inherently advantageous, and believe that it is worthy of publication in its own right. This is particularly very important in the context of unsupervised and exploratory data analysis, which spans a large fraction of our intended applications. In these settings it is important to be able to trust the outcome of your analysis so that you can use it as a basis for further understanding of the system. In contrast, if we were only interested in improving upon a standardized benchmark task, providing a formal approach may matter less --- in that case, we could say “it just works!” However, we and the literature we are building upon are primarily interested in the former and not the latter.
>
> Thus, even though we believe that we do show practical improvements (e.g. in our introduction of novel metrics for convolutional layers, our demonstration that clustering results differ in metric vs non-metric spaces, etc.), we hope the reviewer would still see the value in our paper regardless.
>
> > *"On a related note, I couldn’t tell if the authors claim novelty about the propositions. They are rather elementary, including the generalized version in the supplement, which covers both cases. It wouldn’t surprise me if they appear in a textbook etc although I can’t cite one here. Please clarify this point."*
>
> We do not want to make strong claims of novelty since the proofs are very straightforward. That being said, we have not seen these propositions clearly laid out in any textbook (despite our best efforts to find them!). As we stated in the manuscript, the closest we've seen is chapter 18 of Dryden and Mardia's book. We would happily cite another textbook if the reviewers know of one.
>
> Regardless of whether a similar result might be found in some textbook, it is clear that this result and the rich field of statistical shape analysis is not widely known to machine learning and computational neuroscience researchers. Thus, we believe our paper highlights a number of results and conceptual connections that have been missing from the literature on neural representations for several years, and as such is a valuable contribution.
>
> > *"The experiments would be much more meaningful in the presence of baselines. For instance, if the ABO experiment were repeated, except with non-metric embeddings, but following the same multidimensional scaling and PCA parameters, how different would the embedding of the brain regions be?"*
>
> After looking through the literature we have found a couple instances where papers have used MDS to directly embed non-metric dissimilarities into 2D (Maheswaranathan et al. 2019; Mehrer et al. 2020). We agree it would be useful to perform a baseline comparison to this current practice. Please see our response below for further results, which we will include in the final paper.
>
> We want to clarify that our approach is novel in two respects. First, we use proper metrics to quantify representational dissimilarities. Second, we aim to identify a Euclidean embedding of network representations with low distortion --- i.e. an embedding that accurately reflects the relationships between different networks. This embedding can be used for a wide variety of new analyses --- e.g. PCA, but also classification or regression.
>
> The baseline experiment proposed by the reviewer focuses on the first innovation. However, in this section of the paper we were aiming to emphasize the second innovation. Thus, although one could imagine a new baseline where we use non-metric dissimilarities to find an approximate, high-dimensional Euclidean embedding, we do not believe that this baseline is an obvious approach that someone would have tried without reading our paper first. Or, at the very least, that person would have trouble justifying their approach without having our theoretical framework on hand.
>
> The idea of using Euclidean embeddings as a starting point for downstream tasks is well-motivated in metric spaces --- indeed there is a very old mathematical literature on the topic (Bourgain, 1985) as well as more recent work in machine learning (Vankadara & von Luxburg, 2018). In contrast, there is little precedent for using MDS on non-metric dissimilarities. For example, while there are an agreed upon set of distortion measures for Euclidean embeddings of metric spaces (we look at multiplicative distortion in Fig 4C), it is not clear how one should quantify the quality of the non-metric embeddings and whether one can "trust" these embeddings for the purposes of exploratory data analysis. This might explain why this approach hasn't been followed in past work.
>
> We also want to emphasize that our results could have turned out differently --- it could have turned out that there do not exist Euclidean embeddings of neural network representations that have low distortion. Thus, Fig 4C is a fortunate and novel empirical result that we report here.
>
> > *"Why does it make a big difference to embed in L-dim space first and then perform dim. reduction rather than embedding immediately?"*
>
> These two procedures give very different results, and we think it would strengthen our paper to include a direct comparison. We have done this comparison and drafted a figure that demonstrates this, linked below:
>
> https://pasteboard.co/Ken3UOp.png
>
> Intuitively, multi-dimensional scaling (MDS) with only 2 dimensions produces an embedding of the data with very large distortions in the pairwise distances (see Fig. 4C in our manuscript), so we can't trust that the pairwise relationships / geometry between different networks is faithfully maintained. However, the data can be embedded into L=20 dimensions via MDS with very low distortion. Then, we obtain a 2D view of the data by PCA. This is interpretable as a linear projection reflecting the (nearly exact) geometry of the true metric space. We believe this second approach is clearly superior to what people have previously done.
>
> Looking at the uploaded figure (linked above), we see that the direct MDS procedure onto 2D does vaguely separate the visual cortical areas together as having similar representations. However, the embedded points spread out roughly uniformly on a 2D disk, which is a common outcome when MDS is applied to high-dimensional data. In contrast, our proposed procedure (MDS to 20D, and then PCA to 2D) creates an embedding with more interesting structure. Further, we can better trust that this structure reflects the true properties of the metric space because the original embedding has very low (~5% or less) distortion.
>
> We thank the reviewer for this comment, as we feel it identifies an area of our paper where a small change will help us convey one of our key results much more clearly and persuasively.

---

> > ### Comment · Reviewer_Fmhp · 2021-08-20
> > **mismatch between the problem and the theory**
> >
> > Avoiding a topological perspective significantly limits addressing the identifiability problems neural networks face: in the presence of subsequent deep layers, which are universal function approximators themselves, *any* continuous transformation can potentially be undone, potentially nullifying this development. This is especially true for independently trained neural networks. When the network and dataset are complicated enough to be of practical interest, there will be a mismatch between the assumptions of the underlying model (neural networks) and the math presented in the paper, which could set a misleading precedent. To be clear, the presented propositions and proofs are correct. I just don’t think the rigor of the propositions extend to the problem itself.
> >
> > While it would be ideal to lay out a set of assumptions such that the development of this paper provably avoids such identifiability issues, I think reorienting the presentation to recognize this problem is a bare minimum. (Again, this would be less of an issue if the paper’s claim was mostly on practical results.) Therefore, I decided to keep my score at its original level: I think the paper does have some merit, but the disadvantages outweigh its advantages.
> >
> > I found the authors’ response on the ordering of embedding vs PCA satisfactory.

---

> > > ### Author Response · Authors · 2021-08-20
> > > **No mismatch - these comments misunderstand the current literature**
> > >
> > > Thank you for giving us the opportunity to respond to your further comments. We are glad that you found our additional analysis helpful. Below we add an additional rebuttal regarding the possibility of using topological methods. As indicated in our original rebuttal, we are very happy to modify our exposition to discuss this issue in greater detail.
> > >
> > > -----------------
> > >
> > > > *Avoiding a topological perspective significantly limits addressing the identifiability problems neural networks face: in the presence of subsequent deep layers, which are universal function approximators themselves, *any* continuous transformation can potentially be undone, potentially nullifying this development. When the network and dataset are complicated enough to be of practical interest, there will be a mismatch between the assumptions of the underlying model (neural networks) and the math presented in the paper, which could set a misleading precedent. To be clear, the presented propositions and proofs are correct. I just don’t think the rigor of the propositions extend to the problem itself.*
> > >
> > > **First**, we reiterate that our framework encompasses kernel methods as a special case. Kernel methods are also universal function approximators (Micchelli et al., 2006, https://jmlr.org/papers/v7/micchelli06a.html). Thus, our work in principle extends to the case the reviewer is concerned about. We believe however (see point 2 below) that these nonlinear shape metrics would be most useful when they are highly regularized.
> > >
> > > **Second, and most fundamentally**, the goal of these analyses is ***not*** to find a nonlinear function that perfectly matches two network representations. If one allows for entirely arbitrary alignment functions then every pair of networks will be perfectly aligned, and distance zero from each other! Clearly, the metric space becomes vacuous in this case.
> > >
> > > The principle behind current research is instead the following: if networks learn to exploit similar structure in a dataset (e.g. pulling two classes of images apart into two separate point clouds) then this should be visible or read out by a "simple" mechanism. For example, one can fit linear classifiers to decode image classes from intermediate layers and see that this decoding performance improves as you approach the final layer (e.g., Cohen + Chung et al. 2020, https://doi.org/10.1038/s41467-020-14578-5). Fitting arbitrarily powerful nonlinear decoders would not give the same insight since decoding would be the same across all layers. The large literature on representational similarity analysis that we build upon uses similar reasoning. We provide the following list to demonstrate that this "precedent" of non-topological analysis is already set (and there are still many others in the neuroscience literature such as Gallego et al., 2020, Nature Neuro):
> > >
> > > * Laakso & Cottrell (2010) https://doi.org/10.1080/09515080050002726
> > > * Li et al. (2015) https://arxiv.org/abs/1511.07543
> > > * Morcos et al. (2018) https://arxiv.org/abs/1806.05759
> > > * Raghu et al. (2017) https://arxiv.org/abs/1706.05806
> > > * Kornblith et al. (2019) https://arxiv.org/abs/1905.00414
> > > * Maheswaranathan et al. (2019) https://arxiv.org/abs/1907.08549
> > > * Song & Shmatikov (2019) https://arxiv.org/abs/1905.11742
> > > * Mehrer et al. (2020)  https://doi.org/10.1038/s41467-020-19632-w
> > > * Raghu et al. (2021) https://arxiv.org/abs/2108.08810
> > >
> > > **Third**, we can point to our empirical findings in Figure 5C, which show that linear shape distances are preserved across layers in very deep architectures. That is, if two networks are close together in early layers (e.g. Stem or Stack0), they also tend to be close together in later layers (e.g. Stack2 or AvgPool). The reviewer imagines a "worst-case scenario" where neural representations in early layers are entirely uninformative and randomly reorganized by downstream layers (universal function approximators). This "worst-case scenario" would not actually be a bad outcome from our perspective -- it is useful to know whether or not it happens! Our method provides an answer, and it turns out that a large amount of structure happens to be preserved, even when one uses simple rotational alignments.
> > >
> > > **Fourth**, we reiterate that a topological approach would be the basis for an interesting future project. We do not believe it is fair to critique us for not inventing this particular new approach to study representational similarity. Our work is highly relevant to the field as it stands today (see list of citations above).
> > >
> > > -------------------
> > >
> > > > *While it would be ideal to lay out a set of assumptions such that the development of this paper provably avoids such identifiability issues, I think reorienting the presentation to recognize this problem is a bare minimum.*
> > >
> > > We will add a longer discussion of nonlinear alignments in the final section of our paper, which discusses limitations. This discussion will point to Kornblith et al. (2019, https://arxiv.org/abs/1905.00414), which presents a more detailed argument, backed by empirical experiments, against nonlinear kernel alignments. However, we believe that the utility of nonlinear alignments remains an open question for future research. Again, our framework allows for nonlinear alignments through kernel method extensions, so our work is not predicated on the outcomes of these future investigations.
> > >
> > > -------------------
> > >
> > > > *I found the authors’ response on the ordering of embedding vs PCA satisfactory.*
> > >
> > > Thank you.

---

> > > > ### Comment · Reviewer_Fmhp · 2021-08-23
> > > > **disagreement**
> > > >
> > > > That the present approach is prone to creating vacuous cases in linear or nonlinear alignment scenarios is indeed the point that I’m trying to make. On the other hand, for instance, homological properties of the different representations would remain meaningful. Importantly, computational tools exist for such topological data analysis. Or, for instance, one could study constraining the network architecture to ensure vacuous cases cannot arise. Isometry regularization could potentially be useful, too [Gropp et al, Isometric Autoencoders, 2021].
> > > >
> > > > In a way, my feeling is that this paper attempts to build a strong building on somewhat shaky foundations. While the authors’ argument is that their building is strong, I would be more supportive if they detailed how to modify their presentation to better expose those potentially shaky points.

---

> > > > > ### Author Response · Authors · 2021-08-27
> > > > > **Interesting Discussion**
> > > > >
> > > > > Dear Reviewer Fmhp,
> > > > >
> > > > > Thank you for stimulating this discussion. We have carefully read the Gropp et al (2021) paper and will provide a citation to this in our revised paper alongside a revised exposition along the lines that you suggest. In short, the existing literature focuses on neural "representational *geometry*" whereas you suggest that we move towards studying "representational *topology*." We think this is an intruiging idea, and one that may even fit within our current framework. For example, one could imagine measuring distances based on kernel CCA with radial basis functions --- varying the bandwidth parameter of the kernel can be interpreted as varying the lengthscale over which deformations of the representation manifold are allowed without increasing the generalized shape distance. As we pointed out above, these distances would not be meaningful when the bandwidth parameter is very small or very large. However, you might get meaningful results and structure over a range of intermediate values. This might allow us to generate persistence diagrams that provide insights into neural representations. For a connection between kernel methods and topological data analysis, which may provide a path towards this future approach, see Kusano et al. JMLR 18(189):1−41, 2018.
> > > > >
> > > > > We still believe that a detailed investigation into these possibilities is outside of the scope of our present paper, but that it would improve our manuscript to expand the nonlinear alignment section to highlight these understudied possibilities.
> > > > >
> > > > > Looking forward, we think that representational geometry (using linear/orthogonal alignments) and representational topology (using nonlinear alignments) could provide useful and complementary insights into the structure of neural representations. We believe that our work takes an important first step towards placing these analyses on stronger theoretical grounding.

---

> > > > > > ### Comment · Reviewer_Fmhp · 2021-08-31
> > > > > > **I believe the expanded discussion will significantly increase the impact of the manuscript.**
> > > > > >
> > > > > > I appreciate the authors’ positive response to my critique, and decided to increase my score to 6. I strongly think that an expanded discussion on the topological aspects of the problem will significantly increase the impact of this manuscript.

---

### Official Review · Reviewer_TdwX · 2021-07-16

**Rating:** 9
**Confidence:** 4

**Summary:**

The paper is concerned with issues that are fundamental to basic research on understanding natural and artificial neural networks. Such research often involves measuring the similarity of neural representations (patterns of neural activity). The authors propose desirable attributes for the methods used for such comparisons, e.g. that the similarity function should define a metric metric space: it should satisfy the constraints of equivariance, symmetry and the triangle inequality. They show that many commonly used methods do not satisfy these constraints but can be easily modified to do so. They also construct a metric especially suited to convolutional layers, which traditionally have not been straightforwardly analyzed by existing methods. The authors also discuss practical issues of general importance: They discuss what sample size is required for such metrics to be accurate/robust, which is highly important for neuroscience research where it can be difficult to obtain large sample sizes. They also talk about dimensionality reduction, showing that approximating neural embeddings with low dimensional Euclidean spaces, while suitable for visualization, can highly distort the embedding. They validate their new metrics by showing that it can recover known structure in large scale datasets.

**Limitations And Societal Impact:**

The authors have identified important limitations of their work and highlighted directions for future work. They do not address potential negative societal impact but I can offer no suggestion in that regard. The work does not appear to have any obvious potential negative societal impacts.

**Main Review:**

The work is important, timely, clearly written, and likely to be impactful in both neuroscience and machine learning. It addresses an important issue that I have previously noticed in the work on similarity of neural representations. The analysis appears to be appropriate and correct. The work is clearly motivated and the results actionable. I'm afraid I have nothing constructive to add. My only notes on the paper are positive.

**Time Spent Reviewing:**

1.25

---

> ### Author Response · Authors · 2021-08-10
> **Response**
>
> We thank the reviewer for their positive comments and encouragement. We are very pleased that our paper addresses issues that the reviewer "previously noticed in the work on similarity of neural representations." We hope that this reviewer's enthusiasm helps convince the others that our work fills in important gaps within the current literature. We also appreciate that the reviewer characterizes our contributions as "actionable" suggestions for future work (e.g. our proposal to use approximate Euclidean embeddings for downstream learning tasks).

---

### Official Review · Reviewer_2qso · 2021-07-16

**Rating:** 6
**Confidence:** 3

**Summary:**

The current manuscript presented a unifying framework to analyze the representation similarity between neural networks. The authors first established that the distance metrics between neural representations should satisfy equivalence, symmetry and triangle inequality. And the authors showed that a simply modification of canonical correlation analysis would satisfy these reequipment. Next, the authors identified the transformation matrix that best conform the two representation spaces under various conditions. The current framework revealed the hierarchy of multiple brain areas and visualized the representation structure of a deep neural network.

**Limitations And Societal Impact:**

The authors discussed that limitation of representation analysis and acknowledged that the representation similarity was only loosely related to network function.

**Main Review:**

In the current manuscript, the authors provided theoretical foundation for representation similarity analysis, as well as demonstrated its application in both real and artificial neural networks. While the methods seem to be not novel, the theory was valuable and the application results were intuitive. However, it was confusing that mathematical foundations did not lay out in a way to guide the applications. It seems extra math were provided without giving an example (eg. the permutation invariance, rotation invariance and so on), while the real-world application seems to only relevant to a small fraction of the theory.

**Time Spent Reviewing:**

5

---

> ### Author Response · Authors · 2021-08-10
> **Response**
>
> We thank the reviewer for their thoughtful comments. We will respond to two of their main points.
>
> > *While the methods seem to be not novel, the theory was valuable and the application results were intuitive.*
>
> We thank the reviewer for recognizing the value of our theory to reverse engineering biological and artificial neural networks. While our two main propositions are closely related to concepts in statistical shape analysis (cited throughout our paper), we want to emphasize that these concepts do not appear to be recognized by our target audience, as judged from prior work in this area. Two of our main goals in this work were to (a) introduce key concepts from statistical shape analysis to a broader audience and (b) to extend that theory so it could be applied to the important problem of representational similarity analysis in machine learning and neuroscience.
>
> We develop novel elements of this theory, for example in describing specialized metrics for convolutional hidden layers. Further, in the vast majority of previous shape analysis applications, the shapes are low-dimensional. In contrast, we apply these ideas to high-dimensional neural network representations. This raised a number of new statistical questions that are not relevant to classical shape analysis. For example, how many shape landmarks (corresponding to different network inputs) need to be sampled to get an accurate estimate? We provide an theoretical analysis of sample complexity (see sections 2.5 and the supplement) and we provide an empirical characterization of these issues in fig. 3B-C. In our edits, we will add greater emphasis to these points of novelty in the abstract/intro/conclusion sections.
>
> > *However, it was confusing that mathematical foundations did not lay out in a way to guide the applications. It seems extra math were provided without giving an example (e.g. the permutation invariance, rotation invariance and so on), while the real-world application seems to only relevant to a small fraction of the theory.*
>
> We agree that we could edit the text to better highlight the connections between theory and applications. If the paper is accepted, we will try to condense the theory section and consider moving the paragraph on permutation invariance to the Supplement. We will then use that additional space to motivate the real-world applications and connect them to the theory we developed. We believe that adding a small amount of clarification will go a long way towards resolving the reviewer's criticism.
>
> From reading the totality of reviews, we recognize that we did not make one of our contributions clear enough --- this is the idea that the metric space view of neural representations motivates finding a Euclidean space that well-approximates that metric space, and using those derived Euclidean features for novel downstream analyses (like supervised learning). Such an approach has not been done before, even with non-metric dissimilarities. The second half of the applications section focuses on this contribution, but our exposition did not clearly tie it to the theory developed early on in the paper.
>
> Thus, we wanted to take this opportunity to re-emphasize the novelty of this approach -- e.g. using kernel regression to predict a network's test set accuracy using only the hidden layer representations (Fig. 5D). Our minor edits to the main text will focus on clarifying this contribution. (See also our longer response to Reviewer Fmhp on this point.)

---

### Official Review · Reviewer_2CzF · 2021-07-16

**Rating:** 6
**Confidence:** 3

**Summary:**

This paper proposes a relatively novel method using shape metrics to measure similarity across representations in both neural networks and in the brain, which is an important problem in both fields. The paper compares its method to existing representational metrics and shows that it is able to overcome some of the previous limitations as well as to provide new insights from representations.

**Limitations And Societal Impact:**

Yes.

**Main Review:**

Originality: The incorporation of shape metric into analyzing representation similarity is novel to my knowledge.

Quality: The proposed shape metrics is well motivated and mathematically sound. Given that the authors shows that their method could give different results in hierarchical clustering of representations, which by itself is very interesting, it is unclear how differently it does in other applications comparing to existing methods. The persistent structure across layers, as well as the relationship from test set accuracy and representations might be visible using existing representation similar metrics. I would like to see more experiments done to further validate the method. For example, it can be used to compare representations learned for different tasks to inform multi-task learning, or on a different brain dataset.

Clarity: The paper is clear and well written.

Significance: Application of the method proposed by this paper is very general and extremely useful to both neuroscience and machine learning.

**Time Spent Reviewing:**

4

---

> ### Author Response · Authors · 2021-08-10
> **Response**
>
> We thank the reviewer for their thoughtful comments and suggestions.
>
> The only weakness pointed out by the reviewer is a desire for more experiments and comparisons. While the review process does not allow us to edit the original manuscript to perform new analyses, we will incorporate further experimental results into the Supplement for the camera-ready version (see, e.g., our new baseline analysis provided in our response below to reviewer Fmhp).
>
> We wanted to provide a small number of applications in this paper to demonstrate some new forms of analysis -- e.g. regression on hidden layer representations after removing symmetries (e.g. rotations), which we accomplish by embedding representations into an approximate Euclidean feature space. To our knowledge, previous works have only used entirely unsupervised methods when studying/comparing hidden layer representations (e.g. dimensionality reduction).
>
> We hope that these applications serve as inspiration for more thorough investigations, which we could not fit into this conference paper. We agree that multi-task learning would be a very interesting application of these methods, and we hope to follow up on this in future work. Space permitting, we will include another paragraph in the final conclusion section to outline these future possibilities more fully.

---

> ### Author Response · Authors · 2021-08-20
> **Brief clarification**
>
> Dear Reviewer 2CzF,
>
> We are re-reading the reviews now to respond to Reviewer Fmhp and noticed your following comment that we wish to briefly clarify.
>
> > *Given that **the authors shows that their method could give different results in hierarchical clustering of representations**, which by itself is very interesting, it is unclear how differently it does in other applications comparing to existing method*
>
> This refers to Figure 4B, but it misunderstands the result. **Critically, the two different hierarchical clusterings we show are a difference between our method and existing CCA-based dissimilarity score used in past work.** The CCA-based score has triangle inequality violations, while ours produces a proper metric. The application of hierarchical clustering to non-metric dissimilarities is much less principled and validated than in the metric case (see e.g., Dasgupta & Long, 2005, https://doi.org/10.1016/j.jcss.2004.10.006).
>
> In short, Fig 4B is a concrete example of how we compare our approach to existing work, and show that it produces different results in the setting of unsupervised, exploratory analysis.
>
> Thank you very much for your time and effort in reviewing our paper and we hope that you and the other reviewers can take this clarification into account during your final deliberations.

---

### Decision · Program_Chairs · 2021-09-27

**Decision:**

Accept (Poster)

**Comment:**

This paper introduces a new similarity measure for representations. Given the wide recent use of comparing representations both in analyzing neural networks and in analyzing brain activity. There was enthusiasm from the reviewers about the fact that the metrics introduced are formally based, but however, there was a discussion about whether the paper should include a more formal topological analysis.

A consensus was reached by having the authors discuss their specific assumptions, the potential connections to topological analysis and what could be addressed in future work. Given that the authors have promised to implement these changes, I recommend acceptance.